Mandibular form and function is more disparate in amniotes than in non-amniote tetrapods from the late Palaeozoic

Ponstein Jasper 1 2 3 ponsteij@hu-berlin.de
MacDougall Mark J. 1
http://orcid.org/0000-0002-3824-6992 Schaeffer Joep 4
http://orcid.org/0000-0002-0596-623X Kammerer Christian F. 5
http://orcid.org/0000-0002-2501-9387 Fröbisch Jörg 1 2
1 Museum für Naturkunde Berlin , Berlin , Germany
2 Humboldt-Universität zu Berlin , Berlin , Germany
3 Oertijdmuseum , Boxtel , Netherlands
4 Staatliches Museum für Naturkunde Stuttgart , Stuttgart , Germany
5 North Carolina Museum of Natural Sciences , Raleigh, North Carolina , United States
Bona Paula
Electronic publication date: 2025 Nov 26
Publication date: 2025
Volume: 13
Electronic Location ID: e20243
Received 2025 Apr 16; Accepted 2025 Sep 24
Copyright: © 2025 Ponstein et al.
Copyright year: 2025
Copyright holder: Ponstein et al.
License: This is an open access article distributed under the terms of the Creative Commons Attribution License, which permits unrestricted use, distribution, reproduction and adaptation in any medium and for any purpose provided that it is properly attributed. For attribution, the original author(s), title, publication source (PeerJ) and either DOI or URL of the article must be cited.
License URL: https://creativecommons.org/licenses/by/4.0/

Keywords: Palaeozoic, Tetrapoda, Amniota, Mandible, Disparity, Biomechanics, Morphometrics, Fourier, Evolution, Herbivory

Funding: Elsa-Neumann Stipendium Federal Ministry of Education and Research (BMBF) for the BROMACKER project 01UO2002A Open Access Publication Fund of Humboldt-Universität zu Berlin Jasper Ponstein is funded by an Elsa-Neumann Stipendium. Mark J MacDougall and Jörg Fröbisch are funded by the Federal Ministry of Research, Technology and Space (BMFTR, formerly BMBF) for the BROMACKER project under the grant number: 01UO2002A. The article processing charge was funded by the Open Access Publication Fund of Humboldt-Universität zu Berlin. The funders had no role in study design, data collection and analysis, decision to publish, or preparation of the manuscript.

==============================
Terrestrial tetrapods originated during the Middle Devonian, and the group rapidly diversified throughout the subsequent Carboniferous and Permian periods. Feeding in air rather than water is expected to require changes to tetrapod mandibular form and function. Previous biomechanical studies on jaw evolution, however, found that the increase in functional disparity lagged behind terrestrialisation by approximately 70 Myr, coinciding with the origin of amniotes and herbivory. We expand on a previous dataset composed primarily of non-amniote tetrapods to identify the drivers of this diversification, including representatives of all major amniote clades from the Permo-Carboniferous. First, we measured nine biomechanical traits from 111 tetrapod jaws in medial view. Second, we performed an Elliptical Fourier Analysis on 198 jaws in lateral view and 73 jaws in occlusal view. The first peak in jaw disparity, during the Pennsylvanian, occurs in carnivorous non-amniote tetrapods. However, the jaws of amniotes, particularly those inferred as herbivorous, are consistently more disparate than non-amniote tetrapods from the early Permian, especially in terms of jaw depth, symphysial length and force transmission. Functional and shape disparity of Palaeozoic tetrapod jaws follow a similar pattern that is explained by large-scale faunal turnovers and ecosystem structures.

Introduction

Terrestrial tetrapods originated and diversified into ecosystems during the late Palaeozoic (e.g., Benton et al., 2013). The shift in inhabited medium from water to air necessitated fundamental mechanical, physiological and behavioural changes in tetrapods, including the critical biological function of feeding (e.g., Bramble & Wake, 1985; Lauder, 1985; Schwenk & Rubega, 2005; Markey & Marshall, 2007; Natchev et al., 2015; Heiss, Aerts & Van Wassenbergh, 2018; Van Wassenbergh, 2019; Schwarz et al., 2023). Aquatic gnathostomes notably manipulate water flow relative to the head to capture and swallow food, enabling trophic behaviours such as filter and suction feeding. However, these feeding strategies are ineffective in air, requiring feeding on land to use jaw prehension. To compensate for feeding out of water, Devonian stem-tetrapods such as Tiktaalik roseae adopted an intermediate form of feeding between suction and biting (Lemberg, Daeschler & Shubin, 2021). Moreover, terrestrial tetrapods explored new dietary niches, such as high-fibre herbivory (e.g., King, 1996; Hotton, Olson & Beerbower, 1997; Sues & Reisz, 1998; Reisz & Sues, 2000), insectivory/durophagy (Modesto, Scott & Reisz, 2009; Clack et al., 2019; Mann et al., 2023) and macropredation targeting other tetrapods (e.g., Van Valkenburgh & Jenkins, 2002; Brocklehurst, 2019; Singh et al., 2024) during the subsequent Carboniferous and Permian periods.

It is thus expected that the mandible, a structure that is specialised in feeding, would be subject to substantial change over the fish-tetrapod transition. However, the acquisition of novel mandibular characters along the tetrapod stem appears to have been a rather slow process throughout the Devonian and Early Carboniferous (Ahlberg & Clack, 1998). Anderson, Friedman & Ruta (2013) found that the functional diversification of the tetrapod mandible lagged behind the evolution of terrestrialisation for about 70–80 Myr. Following this long period of functional stability, there was a sharp increase in jaw disparity during the Late Carboniferous to early Permian (Anderson, Friedman & Ruta, 2013; Neenan et al., 2014). This increased rate of change coincides with the advent of both amniotes and herbivores (Anderson, Friedman & Ruta, 2013). However, as Anderson, Friedman & Ruta (2013) focused primarily on the fish-tetrapod transition, their taxon sample included few amniotes. Notably, their selection of amniote taxa excluded major late Palaeozoic groups such as “parareptiles”, captorhinids and therapsids. It is still unclear what caused the delay and what eventually drove the functional diversification of tetrapod jaws during the Palaeozoic.

Two major functional differences between amniotes and non-amniote tetrapods are hypothesised to have lifted constraints on the skull, allowing amniotes to evolve more divergent craniomandibular shapes. Firstly, restructuring of the jaw adductor musculature provided amniotes with a static-pressure bite, allowing them to exert pressure when the mouth is closed and exhibit greater biting control and dental occlusion (Carroll, 1969; Janis & Keller, 2001; Reisz, 2006). This change facilitated more mechanically efficient jaws, powering greater biting forces, particularly at the back of the jaw. Thus, it is expected that amniotes can reach higher posterior mechanical advantages than non-amniotes. Secondly, non-amniote tetrapod jaw shape is constrained by their reliance on buccal pumping (Janis & Keller, 2001). This mode of respiration, involving rapid oscillation of the buccal floor, represents the ancestral mode of breathing in tetrapods (e.g., Brainerd, Ditelberg & Bramble, 1993; Brainerd, 2016; Brainerd, 1999; Brainerd & Owerkowicz, 2006). This is correlated with a skull that is dorsoventrally flattened, with a high width-to-length ratio (Szarski, 1962; Janis & Keller, 2001) and accordingly, a dorsoventrally flattened mandible. We expect that amniotes have the ability to acquire deeper jaws than non-amniote tetrapods. The symphysis in non-amniote tetrapods is expected to be short, as blunt mandibles generate greater negative pressure (Werth, 2006; Vicari, Boccone & Pandolfi, 2024). Amniotes developed costal aspiration alongside buccal pumping (Brainerd, Ditelberg & Bramble, 1993; Brainerd & Owerkowicz, 2006), releasing these constraints on skull shape (Janis & Keller, 2001). Together, the innovations of a static-pressure bite and costal aspiration could explain the acceleration in jaw disparity coinciding with the diversification of amniotes, following a lengthy period of relative stasis in non-amniote tetrapods.

Additionally, new dietary niches often result in adaptive radiations. Herbivores in particular are known to occupy different regions of functional morphospace than their non-herbivorous relatives in such disparate gnathostome clades as actinopterygians (Bellwood, 2003) and squamates (Stayton, 2006). The evolution of grazing in actinopterygians during the Eocene co-occurred with a massive expansion of actinopterygian jaw functional morphospace, introducing taxa with short, force-optimised jaws (Bellwood, 2003). Stayton (2006) found that the jaws of herbivorous squamates converge in areas of functional morphospace that are unoccupied by non-herbivorous taxa. Within modern gnathostome clades, herbivores typically have more efficient force transmission than non-herbivorous relatives (e.g., Maynard Smith & Savage, 1959; Stayton, 2006; Navalón et al., 2018; Grossnickle, 2020; Ma et al., 2020; Ponstein et al., 2024). Berks et al. (2025) recently suggested that the evolution of herbivory in amniotes and diadectomorphs relaxed constraints on jaw function, thereby causing the observed increase in tetrapod jaw disparity in the Late Carboniferous. These authors added that other biological factors might have played a role in lifting constraints on jaw shape, which could have preceded herbivory. Hence, the causal relation between the origin of herbivory and rise in jaw disparity in Palaeozoic tetrapods remains unclear.

This study has three primary aims: (1) Expand on the mandibular dataset used by Anderson, Friedman & Ruta (2013), filling in taxonomic gaps and charting disparity patterns into the subsequent Permian period, (2) compare patterns of functional and morphological disparity in Permo-Carboniferous amniote and non-amniote tetrapods, and (3) identify the drivers of tetrapod jaw diversification in the late Palaeozoic.

Materials and Methods

Selection of specimens

We collected images of mandibles of 202 tetrapod taxa spanning the Carboniferous and Permian periods from the literature, specimen photographs and 3D-data (computed tomographic and surface scans). According to the “traditional” cladistic definition of Amniota (i.e., the amniote crown group; Gauthier et al., 1989, but see discussion below), our sample includes 85 non-amniote tetrapods and 117 amniotes (83 synapsids and 34 sauropsids). Of these mandibles, 198 are figured in lateral view, 111 are figured in medial view and 73 are figured in occlusal view. We aimed to obtain a taxonomically diverse dataset in each time bin. These time bins correspond to the stages of the Carboniferous and Permian periods (cf. Cohen et al., 2023). We were unable to obtain suitable tetrapod jaw specimens from the Tournaisian, and so our sample covers the Viséan-Changhsingian (approximately 105 Myr). For a list of taxa and specimens included, see File S1.

This study aims to compare jaw disparity in amniotes and non-amniote tetrapods. However, the precise contents of Amniota remain disputed (e.g., Marjanović & Laurin, 2019; Modesto, 2024). As such, when comparing amniotes to non-amniote tetrapods, we employ four different definitions of Amniota based on published phylogenetic hypotheses. Hypothesis 1, the “traditional” cladistic hypothesis (e.g., Gauthier et al., 1989; Laurin & Reisz, 1995), places recumbirostran microsaurs and diadectomorphs as non-amniote tetrapods, whereas Captorhinidae, Araeoscelidia and Protorothyrididae are classified as crown amniotes. This is used as the baseline hypothesis. Hypothesis 2 includes Diadectomorpha as basal synapsids within Amniota (e.g., Berman, 2013; Klembara et al., 2019, 2021; Clack, Smithson & Ruta, 2022; Ponstein, MacDougall & Fröbisch, 2024). Hypothesis 3 treats Captorhinidae, Araeoscelidia and Protorothyrididae as non-amniote tetrapods (e.g., Simões et al., 2022; Klembara et al., 2023; Jenkins et al., 2024, 2025). Lastly, hypothesis 4 places recumbirostran microsaurs as sauropsid amniotes (e.g., Pardo et al., 2017; Mann & Maddin, 2019; Mann, Pardo & Maddin, 2022, Mann, Pardo & Sues, 2023). In addition, there is some disagreement on the precise contents of the two major amniote subclades, Synapsida and Sauropsida. For consistency among the four hypotheses, we consider Varanopidae as synapsids (e.g., Angielczyk & Kammerer, 2018; but see Ford & Benson, 2018, 2020 for alternative interpretations). Finally, the taxon Alveusdectes fenestratus (Liu & Bever, 2015) is classified as a synapsid, but not a diadectid (Spindler, Werneburg & Schneider, 2019; Ponstein, MacDougall & Fröbisch, 2024), in all analyses. For a taxonomic breakdown of each phylogenetic hypothesis, see Table 1.

Table 1 Taxonomic breakdown of dataset.

Phylogenetic hypothesis	Non-amniote tetrapods	Synapsids	Sauropsids	
H1	85	83	34	
H2	78	90	34	
H3	100	83	19	
H4	75	83	44	

We also test the effect of an herbivorous diet on mandibular disparity, as herbivores are often documented to expand the craniomandibular morphospace occupation of an ancestrally carnivorous clade. We follow the dietary classifications of Hellert et al. (2023) for synapsid herbivores, and add the therocephalian Purlovia maxima following the dietary hypotheses of Ivakhnenko (2011). All members of Diadectidae, Bolosauridae and Pareiasauria are classified as herbivores (e.g., Hotton, Olson & Beerbower, 1997; Reisz & Sues, 2000). Within Captorhinidae, only taxa with multiple tooth rows capable of palinal jaw motion are considered herbivorous (cf. Brocklehurst, 2017). The classification criteria include dental and postcranial morphology that do not overlap with our functional measurements. All remaining taxa are treated as non-herbivores. Using these definitions, our dataset consists of 56 herbivores and 146 non-herbivores. For the assigned dietary categories, see File S1.

Biomechanical measurements

We took nine biomechanical measurements from the medial side of the jaw (Table 2; File S2). These measurements are widely applied to study jaw function and evolution in diverse gnathostome clades (e.g., Bellwood, 2003; Anderson, 2009; Anderson et al., 2011; Anderson, Friedman & Ruta, 2013; Stubbs et al., 2013; Button, Rayfield & Barrett, 2014; Stubbs & Benton, 2016; MacLaren et al., 2017; Benevento, Benson & Friedman, 2019; Ma et al., 2020; Singh et al., 2021, 2024; Foffa, Young & Brusatte, 2024; Ponstein et al., 2024). We chose to omit characters concerning individual teeth, as several of the dicynodont therapsid taxa included are edentulous. The rationale and procedure for each measurement is discussed in File S2.

Table 2 Functional measurements used in this study.

Measurements	
1. Anterior mechanical advantage	
2. Posterior mechanical advantage	
3. Opening mechanical advantage	
4. Maximum aspect ratio	
5. Average aspect ratio	
6. Articular offset	
7. Relative ‘dental row’ length	
8. Relative adductor fossa length	
9. Relative symphysial length	

All measurements were done using the GNU Image Manipulation Program (GIMP 2.10.36; The GIMP Development Team, 2023). The raw measurement data (File S4) were transferred into the statistical computing environment R (R Core Team, 2021). From these data, the distributions of functional measurement values were visualised in box plots, comparing amniote with non-amniote taxa. We then tested if group means are statistically significantly different by applying Bonferroni-corrected pairwise t-tests, using the pairwise.t.test function of the stats package (R Core Team, 2021). We also tested whether amniote variance is statistically significantly different from non-amniote tetrapod variance by applying Levene’s Test for Equality of Variance, using the leveneTest function of the car package (Fox et al., 2019). To test the effect of new dietary niches on functional disparity, we also compared herbivores with non-herbivorous tetrapods.

Each measurement was then standardized by Z-transformation so that the average value for each trait approaches 0 and the standard deviation approaches 1, ensuring equal weights are given to the individual measurements (e.g., Benevento, Benson & Friedman, 2019). The measurement data were subsequently subjected to Principal Component Analysis (PCA) using the prcomp function of the stats package (R Core Team, 2021). We assessed disparity and centroid position separation for the aforementioned clade and dietary categories. We also compared between the two major subclades of Amniota: Synapsida and Sauropsida. To assess disparity, we calculated the sum of variances from the PCA coordinates, using the dispRity function of the dispRity package (Guillerme, 2018). Sum of variances characterizes the volume that a subgroup occupies in morphospace (Guillerme et al., 2020). This disparity metric is relatively insensitive to small sample size effects (Wills, 1998; Brusatte et al., 2008; Hopkins & Gerber, 2017). Significance of morphospace separation is tested based on the centroid position on all PC axes through PERMANOVA tests using the vegan package (Oksanen et al., 2024).

Morphological analysis

Elliptical Fourier analysis (EFA), often referred to as outline analysis, is increasingly being used to study shape in vertebrate mandibles (e.g., Schmittbuhl et al., 2007; Hill et al., 2018; Navarro, Martin-Silverstone & Stubbs, 2018; Schaeffer et al., 2020; Deakin et al., 2022). The method provides a detailed approximation of a given shape by fitting a series of harmonics onto the specimen image (e.g., Kuhl & Giardina, 1982; Caple, Byrd & Stephan, 2017). EFA utilizes four coefficients to define each harmonic as an ellipse (Caple, Byrd & Stephan, 2017). These coefficients can then be used to compare different shapes. EFA is an effective approach for studying a taxonomically diverse dataset as it does not depend on homologous landmarks.

We collected 198 figures of Palaeozoic tetrapod jaws in lateral view (File S1). The specimen images were converted to silhouettes, excluding the dentition, using Adobe Photoshop CS6 (Adobe Inc., San Jose, CA, USA). All jaws are figured in left lateral view. Specimens of which only the right hemimandible is preserved, or specimens of which the right hemimandible is more complete than the left hemimandible, are mirrored. When necessary, minor reconstructions to biologically insignificant areas of the jaws were carried out in Adobe Photoshop.

Previous studies on 2D jaw shape only considered lateral view (e.g., Hill et al., 2018; Navarro, Martin-Silverstone & Stubbs, 2018; Schaeffer et al., 2020; Berks et al., 2025). However, not all relevant aspects of jaw anatomy are captured in lateral view. Notably, the geometry of the jaw arch in occlusal view varies from a wide U-shape in non-amniote tetrapods to a narrow V-shape in amniotes. In addition, some variation pertains to the widening of the jaw ramus, such as the presence of a dentary shelf in moradisaurine captorhinids (e.g., Ricqlès & Taquet, 1982; Modesto et al., 2019), which is an adaptation related to herbivory. To appreciate the full extent of morphological variation in the dataset, we additionally analyse the jaws in occlusal view. As a silhouette of a jaw in occlusal view is indistinguishable from a silhouette in ventral view, we also considered figures of occluded jaws in ventral view. We collected 73 images of jaws in occlusal or ventral view (File S1), and converted these to silhouettes using Adobe Photoshop.

These silhouettes were read in R through the package Momocs (v. 1.1.6; Bonhomme et al., 2014) and were converted into digitised outline closed curves. Jaws in occlusal view were split at the symphysis and only the silhouette of one jaw ramus was considered, as the Momocs package could not read the complete jaw arch properly. As in the lateral jaw dataset, the occlusal jaw is figured as the left hemimandible. For both the lateral and occlusal analyses, all jaw rami were then centred, scaled and aligned to remove the noise resulting from differences in size and orientation. We applied a set of 660 x and y-coordinate points to the outline profile of the jaws in lateral view and a set of 500 x and y-coordinate points to the jaw rami in occlusal view. The number of harmonics required to account for the shape variation was calibrated using the calibrate_harmonicpower command. The Fourier coefficient data accounting for 99% of the shape variation in the original jaws or jaw rami were subjected to principal component analysis using the PCA function.

Similar to what is done for the functional dataset, we compared the disparity and centroid positions of amniotes vs. non-amniotes, Synapsida vs. Sauropsida, and herbivores vs. non-herbivores for both shape datasets.

Temporal trends

To analyse functional and morphological disparity through time, we subdivide the Carboniferous and Permian periods into fifteen time bins corresponding to the sampled geological stages (Viséan-Changhsingian; cf. Cohen et al., 2023). Taxa were assigned to time bins that fall inside their First Appearance Datum (FAD) to Last Appearance Datum (LAD) range. FAD and LAD data were imported from the Paleobiology Database (Uhen et al., 2023; www.paleobiodb.org), unless otherwise specified (File S1). These data were used to generate plots showing the number of specimens included per time bin. Each time bin includes at least five taxa. Then, for both the functional and the morphological analysis, sum of variance is calculated for the subset of taxa in each time bin across all PC axes, and plotted per time bin. Disparity time-series of mandibles in occlusal view is analysed through five geological series (Mississippian, Pennsylvanian, Cisuralian, Guadalupian, Lopingian), as the Moscovian, Kasimovian and Gzhelian stages each contain only a single jaw in occlusal view. For both the functional and shape morphospace, we plot subsets of morphospace corresponding to each geological series to help visualise large-scale patterns. Lastly, to investigate the relative contribution of each major group, i.e., “non-amniote tetrapods”, Synapsida and Sauropsida, to the total disparity, we calculate partial disparity over time. Partial disparity of a group is defined as the average distance to the centroid, divided by number of included specimens minus 1 (Foote, 1993). This is done for every time bin in base R.

All the code and relevant files are made available in File S7.

Results

Jaw biomechanics

The jaws of amniotes are more diverse biomechanically than non-amniote tetrapods. Under the baseline phylogenetic hypothesis, amniotes achieve greater variation in five out of nine biomechanical measurements; anterior and posterior mechanical advantage, articular offset, adductor fossa length, and symphysial length (Fig. 1; File S3A). Intriguingly, amniotes show the greatest dissimilarity to non-amniotes when captorhinids, araeoscelidians and protorothyridids are placed outside of Amniota. Under this hypothesis, the range of amniotes is higher in all traits but tooth row length (Fig. 1; File S3A). Amniotes score higher than non-amniotes on opening mechanical advantage, maximum aspect ratio, average aspect ratio, articular offset and symphysial length regardless of how Amniota is defined (Fig. 1; File S3A). Anterior mechanical advantage is significantly higher than in non-amniote tetrapods when either Diadectomorpha or Recumbirostra are included as amniotes (File S5). Non-amniote tetrapods have significantly longer tooth rows and longer adductor fossae than amniotes under all four phylogenetic hypothesis (Fig. 1, File S3A).

Figure 1 Box plots of all nine functional characters, comparing amniotes to non-amniotes under the baseline phylogenetic hypothesis (see Methods) and herbivores to non-herbivores.

(A) Anterior mechanical advantage, (B) Posterior mechanical advantage, (C) Opening mechanical advantage, (D) Maximum aspect ratio, (E) Average aspect ratio, (F) Articular offset, (G) Relative “dental” row length, (H) Relative adductor fossa length, (I) Relative symphysial length. Black bar within box plots represents the median values, and boxes themselves indicate the interquartile range.

The functional traits of herbivores are vastly different from those of non-herbivorous tetrapods (Fig. 1). Herbivores have higher values for all mechanical advantages (anterior, p < 0.01; posterior, p = 0.015, opening, p < 0.01), aspect ratios (average, p < 0.01, maximum, p < 0.01), articular offset (p < 0.01) and symphysial length (p < 0.01) than non-herbivores (File S3A).

An overview of the minima and maxima for each measurement is presented in Table 3, while the complete functional dataset can be found in File S4. The lowest anterior mechanical advantage is found among derived theriodont therapsids, such as Dvinia prima (0.080) Annatherapsidus petri (0.086), Procynosuchus delaharpeae (0.087) and Cynariops robustus (0.14). The lowest value among non-therapsids is measured in the younginid neodiapsid Youngina capensis (0.15). Among the highest anterior mechanical advantage values measured are herbivorous taxa of different clades—the venyukovioid anomodont Ulemica invisa (0.29), the diadectid diadectomorph Diadectes lentus (0.33), the tapinocephalid dinocephalian Ulemosaurus svijagensis (0.32) and the captorhinid Rhodotheratus parvus (0.32). With regards to posterior mechanical advantage, both the herbivorous edaphosaurid E. boanerges and the diadectid D. lentus are plotted as positive outliers among the amniote and non-amniote samples, respectively, indicating extraordinary force transmission for posterior biting in these taxa. Among the lowest values for posterior mechanical advantage are, similar to the anterior mechanical advantage, derived therapsids (i.e., therocephalians and gorgonopsians), due to their reduced adductor fossae. Anterior and posterior mechanical advantage are significantly but weakly correlated (p = 2.8 × 10−16; R2 = 0.46). Dicynodont anomodonts score particularly high on opening mechanical advantage; Daptocephalus leoniceps (0.20), Oudenodon bainii (0.22) and Niassodon mfumukasi (0.23) are consistently plotted as positive outliers. Low values are found among such carnivorous tetrapods as the scylacosaurid therocephalian Glanosuchus macrops (0.031), the eryopid temnospondyl Onchiodon labyrinthicus (0.031), the chroniosuchid chroniosuchian Chroniosaurus dongusensis (0.032) and the gephyrostegid Gephyrostegus bohemicus (0.034). High values of the maximum aspect ratio metric are dominated by “parareptiles”; pareiasaurs such as Deltavjatia rossica (0.49) with a deep angular boss and bolosaurids such as Bolosaurus major (0.42) with a tall coronoid process. Average and maximum aspect ratio are both significantly and strongly correlated (p = 2.2 × 10−16; R2 = 0.79). Higher articular offset values are typically achieved by herbivorous tetrapods such as the dicynodont anomodonts N. mfumukasi (0.33) and Eosimops newtoni (0.27), the edaphosaurid E. boanerges and the diadectid D. lentus (0.26), with the exception of the carnivorous diplocaulid Diploceraspis burkei achieving the highest value (0.38). Relative tooth row length ranges from 0.25 (Oestocephalus amphiuminus) to 0.69 (Pantylus cordatus) (Table 3). Within definitive amniotes, this ranges from 0.29 (G. macrops) to 0.63 (P. delaharpeae). As for the jaw closing mechanical advantage, relative adductor fossa length is lowest in derived therapsids. This includes the cynodont D. prima (0.12), the dicynodont anomodont O. bainii (0.010) and the therocephalian Karenites ornamentatus (0.10). The adductor fossa length of captorhinids like C. aguti (0.39), R. parvus (0.37) and Labidosaurus hamatus (0.34) is among the longest of the sampled tetrapods. Finally, long relative symphysial lengths are achieved by dicynodont anomodonts, such as Aulacephalodon kapoliwacela (0.39), Euptychognathus bathyrhynchus (0.26) and D. leoniceps (0.26), as well as other herbivorous synapsids like the tapinocephalid dinocephalian Moschognathus whaitsi (0.21) and the edaphosaurid E. boanerges (0.21). Non-amniote tetrapods commonly have narrow symphyses, such as the temnospondyls Dendrerpeton helogenes (0.020), O. labyrinthicus (0.021) and the anthracosaur Anthracosaurus russelli (0.026).

Table 3 Overview of minimum and maximum values for each measurement.

Measurement	Lowest value	Taxon	Highest value	Taxon	
Anterior MA	0.0796	Dvinia prima	0.358	Casea broilii	
Posterior MA	0.130	Annatherapsidus petri	1.126	Edaphosaurus boanerges	
Opening MA	0.0314	Glanosuchus macrops	0.229	Niassodon mfumukasi	
Maximum AR	0.119	Youngina capensis	0.556	Scutosaurus sp.	
Average AR	0.0778	Lethiscus stocki	0.312	Daptocephalus leoniceps	
Articular offset	0.00368	Chroniosaurus dongusensis	0.377	Diploceraspis burkei	
Dental row length	0.247	Oestocephalus amphiuminus	0.693	Pantylus cordatus	
Adductor fossa length	0.0676	Dvinia prima	0.389	Captorhinus laticeps	
Symphysial length	0.0188	Claudiosaurus germaini	0.390	Aulacephalodon kapoliwacela	
Note:

MA, Mechanical Advantage; AR, Aspect Ratio.

Our Principal Component Analysis (Figs. 2A, 2C, 2E) captures the majority of the variation in the lower jaw of Palaeozoic tetrapods. The first two PC axes combined explain 60.58% of variation (File S3B). PC1 accounts for 37.61% of the variation, whereby positive PC1 values are associated with high opening mechanical advantage, maximum aspect ratio, average aspect ratio, articular offset and symphysial length (eigenvector coefficients > 0.35). PC2 (22.97%) is strongly negatively correlated with anterior mechanical advantage, posterior mechanical advantage and relative adductor fossa length (eigenvector coefficients < −0.50). PC3 is the lowest PC axis that explains more than 10% (13.94%). This axis is strongly correlated with relative tooth row length (eigenvector coefficient = 0.80).

Figure 2 PCA and disparity of jaw biomechanics, under the baseline phylogenetic hypothesis (see Methods).

(A, B) Comparing amniotes with non-amniote tetrapods, (C, D) comparing synapsids with sauropsids, (E, F) comparing herbivores with non-herbivores. A. ohioensis = Adamanterpeton ohioensis; A. petri = Annatherapsidus petri; C broilii = Casea broilii; C. dongusensis = Chroniosaurus dongusensis; D. burkei = Diploceraspis burkei; D. hesperis = Desmatodon hesperis; D. lentus = Diadectes lentus; D. leoniceps = Daptocephalus leoniceps; D. prima = Dvinia prima; E. boanerges = Edaphosaurus boanerges; Y. capensis = Youngina capensis.

Amniotes occupy a greater portion of functional morphospace than non-amniote tetrapods (p = 0 for all definitions of Amniota) (Fig. 2B; File S3D). In addition, the centroids of amniotes and non-amniote tetrapods are significantly different (p = 0.001 for all definitions of Amniota). Each of the major amniote subclades, Synapsida and Sauropsida, individually occupy a greater portion of morphospace than non-amniotes regardless of the phylogenetic hypothesis (Fig. 2D). Synapsids do occupy a greater region than sauropsids under all phylogenetic hypotheses (p = 0), but the centroid positions are only significantly different under phylogenetic hypothesis 1 (F = 2.99; p = 0.039) and 4 (F = 3.75; p = 0.012) (File S3D). Considering diet, herbivores inhabit a greater region of morphospace than non-herbivorous tetrapods (W = 999360; p = 0) and the centroids between the two dietary categories are significantly different (F = 25.70; p = 0.001) (Figs. 2E, 2F; File S3D).

PC axes 1 and 2 exemplify the greater functional disparity of amniotes with respect to non-amniote tetrapods (Fig. 2A). The highest PC1 value is found in the dicynodont anomodont Daptocephalus leoniceps (5.8) (File S3C). This corresponds to a deep jaw with a relatively slow jaw opening, whereby the articular sits at a high angle with respect to the functional “dental” row. The three highest values are achieved by dicynodont anomodonts. High PC1 values in general are dominated by herbivorous taxa such as the edaphosaurid Edaphosaurus boanerges (4.01), the diadectid Diadectes lentus (4.01), the dinocephalian Ulemosaurus svijagensis (3.64) and the pareiasaur Scutosaurus sp. (3.29) (Fig. 2A; File S3C). Two carnivorous tetrapods have a PC1 value greater than 1.9; the diplocaulid Diploceraspis burkei (2.56) and the sphenacodontid Dimetrodon sp. (1.96). The majority of non-amniote tetrapods are confined between PC1 values of −3.0 and 1.0, depending on the definition of Amniota used (Fig. 2A; File S3C). Positive non-amniote tetrapod outliers in PC1 include the diadectids D. lentus and Desmatodon hesperis (excluding hypothesis 2) and the diplocaulid D. burkei, reflecting their relatively deep jaw rami and, especially in the case of D. burkei, high articular offset. The chroniosuchian Chroniosaurus dongusensis has the lowest PC1 value in our dataset (−2.83), followed by other carnivorous taxa such as the younginid Youngina capensis (−2.82), Doragnathus woodi (−2.64) and Varanosaurus acutirostris (−2.37) (File S3C). Regarding PC2, the highest value is recorded in the cynodont Dvinia prima (4.27). Only derived therapsids, such as dicynodonts, gorgonopsians and therocephalians, inhabit PC2 values exceeding 1.5 (Fig. 2A; File S3C), reflecting jaws with a relatively short insertion site of the adductor musculature and thereby resulting in low transmission of adductor muscle force to bite force. Among the lowest PC2 values are various herbivorous tetrapods, such as the caseid Casea broilii (−3.28), the diadectid D. lentus (−2.98) and the edaphosaurid E. boanerges (−2.81) (Fig. 2E; File S3C).

Disparity of tetrapod jaw mechanics varies greatly throughout the late Palaeozoic (Fig. 3A). Throughout the Mississippian, disparity is low. The sampled taxa in this interval, all non-amniote tetrapods, are confined to negative PC1 and PC2 values (Fig. 4A). A first peak in disparity appears during the Moscovian stage (Fig. 3A). This corresponds to an excursion into both positive PC1 and PC2 values in the Pennsylvanian (Fig. 4A). A slight drop in the Kasimovian directly follows the Moscovian peak. Although the first amniotes appear during the Pennsylvanian, their relative contribution to the total disparity during this interval remained less than 50%, except when diadectomorphs are included as amniotes (Fig. 3C). From the Gzhelian, disparity steadily increases until it reaches a second peak during the Sakmarian (Fig. 3A). Sampled taxa from the Cisuralian cover nearly the entire breadth of PC1. Several carnivores, i.e., Dimetrodon sp., Diploceraspis burkei and Delorhynchus cifellii exhibit positive PC1 scores, so there is significant overlap between herbivores and non-herbivores. During the remainder of the early Permian, disparity gradually decreases until the Roadian (Fig. 3A). Despite this, the relative contribution of amniotes steadily increases from the Artinskian onwards (Fig. 3C). Total disparity experiences the highest peak in the Capitanian (Fig. 3A). During the Guadalupian, taxonomic diversity in the bottom left quadrant reduces, and more taxa with either high PC1 or PC2 values evolve (Fig. 4A). From this interval, the overlap in morphospace between herbivorous taxa and non-herbivores decreases, as only herbivores achieve PC1 scores greater than 2.0. By the Lopingian, the bottom right quadrant is devoid of taxa and morphospace occupation is generally less dense (Fig. 4A). This is correlated with a sharp drop in disparity into the Wuchiapingian and Changhsingian stages (Fig. 3A).

Figure 3 Jaw biomechanics disparity through time, under the baseline phylogenetic hypothesis.

(A) Disparity per time, (B) number of specimens per time bin with the thick line denoting all specimens, fair line amniotes and dashed line non-amniote tetrapods, (C) partial disparity for each major group over time. RFC = Rainforest Collapse, OE = Olson’s Extinction, GE = Guadalupian Extinction. Light goldenrod denotes non-amniote tetrapods, blue denotes synapsids and green denotes sauropsids. Representative silhouettes: Crassigyrinus scoticus for non-amniote tetrapods, Lycosuchus vanderrieti for synapsids and Belebey vegrandis for sauropsids. Specimen sources in File S1.

Figure 4 Evolution of morphospace occupation (PC1 vs. PC2) per time bin (epoch), under the baseline phylogenetic hypothesis.

(A) jaw biomechanics; (B) lateral shape analysis; (C) occlusal shape analysis. Guad., Guadalupian; Lop., Lopingian. Light goldenrod circles denote non-amniote tetrapods, green squares denote sauropsids, blue squares denotes synapsids.

Lateral shape morphospace

The lateral outline of Palaeozoic tetrapod jaws is well explained by the first two PC axes (58.86%) (File S3F). The first ten axes describe over 95% of variance. From PC axis 4, each individual axis describes less than 7%. There is a total of 40 PC axes (File S3F).

PC axis 1 (44.55%) describes dorsoventrally shallow vs. deep jaws, with higher values corresponding to deeper jaws (File S5). PC axis 2 accounts for 14.31% of the variation. Jaws at high PC2 values are generally horizontal and featureless. On the other hand, at negative PC2 values jaws are increasingly more complex, with clearly defined post-coronoid regions and a narrow dentary ramus that curves dorsally towards the symphysis (File S5).

Amniotes are more disparate and occupy significantly different regions of morphospace than non-amniote tetrapods, regardless of which definition of Amniota is used (Fig. 5B; File S3G). Additionally, under each phylogenetic hypothesis, both subclades of Amniota are more disparate than non-amniote tetrapods (File S3G). Sum of variance of sauropsid morphospace occupation is greater than that of synapsids, except when recumbirostrans are included in Sauropsida (File S3G). The centroids of both groups are significantly different, except when captorhinids, araeoscelidians and protorothyridids are removed from Sauropsida (File S3G). Herbivores are vastly more diverse than non-herbivores (Fig. 5F). The centroid position of herbivores is significantly different than that of non-herbivores (F = 54.96; p = 0.001). In addition, herbivore morphospace occupation is much greater (W = 996190; p = 0).

Figure 5 PCA and disparity of the lateral shape analysis, under the baseline phylogenetic hypothesis.

(A, B) Comparing amniotes with non-amniote tetrapods, (C, D) comparing synapsids with sauropsids, (E, F) comparing herbivores with non-herbivores. B. qilianicus = Biseridens qilianicus; B. striatus = Bolosaurus striatus; D. lentus = Diadectes lentus; L. stocki = Lethiscus stocki; M. tenuidens = Mesosaurus tenuidens; O. mayorum = Orovenator mayorum; O. whaitsi = Odontocyclops whaitsi; P. attheyi = Pholiderpeton attheyi; S. multidentata = Sigournea multidentata; S. progressus = Sauroctonus progressus.

The highest PC1 values, corresponding to deep jaw rami, are found exclusively in herbivorous tetrapods (Fig. 5E). This includes the pareiasaur Scutosaurus sp. (0.33), diadectid diadectomorph Diadectes lentus (0.29) and the dicynodont anomodont Odontocyclops whaitsi (0.28) (File S3F). In addition, the lowest values are occupied by the diapsid Orovenator mayorum (−0.27) and the mesosaurid Mesosaurus tenuidens (−0.26), which are characterised by dorsoventrally narrow or relatively elongate jaws. Non-amniote tetrapods are largely confined to slender jaws, exemplified by PC1 values ranging from −0.22 (Lethiscus stocki) to 0.11 (Diploceraspis burkei), with the diadectids Diadectes lentus and Desmatodon hesperis (0.18) as notable outliers. Negative PC2 values below −0.14 are almost exclusively held by therapsid synapsids (File S3F), representing a complex bipartite jaw. The lowest value is measured for the gorgonopsian Sauroctonus progressus (−0.42), accompanied by the gorgonopsian Cynariops robustus (−0.33) and the therocephalian Glanosuchus macrops (−0.27). The previously mentioned well-developed post-coronoid region refers to the reduced postdentary bones in derived therapsids, leading to this bipartite jaw morphology. As such, synapsids—and in particular therapsids—achieve the greatest breadth in PC2. However, the bolosaurids Bolosaurus striatus (−0.22) and B. major (−0.19) are also plotted at negative PC2 values (File S3F) by virtue of their tall coronoid processes. On the other side, the highest PC2 values are obtained in the pareiasaur Scutosaurus sp. (0.26). Apart from the angular boss, this jaw is interpreted as a featureless horizontal beam. The non-amniote tetrapod sample is limited between PC2 values of 0.077 (Batropetes palatinus) and -0.137 (“Lysorophus” sp.). When recumbirostrans are considered as amniotes, the non-amniote sample is limited even further to values between 0.067 (Sigournea multidentata) and −0.081 (Pholiderpeton attheyi) (File S3F).

Temporally, shape disparity in lateral view follows an overall similar pattern to that of the functional disparity. During the Mississippian, shape morphospace of the jaw in lateral view is confined to low and negative PC1 values (Fig. 4B). This is reflected in a low overall disparity (Fig. 6A). Throughout the Pennsylvanian, jaw shape disparity in lateral view increases gradually starting from the Moscovian and peaks into the Gzhelian (Fig. 6A). Both amniotes and non-amniote tetrapods expand into higher PC1 values during the Pennsylvanian (Fig. 4B), corresponding to deeper jaw rami. The highest PC1 value is achieved by the herbivorous diadectid Desmatodon hesperis. Breadth of PC2 occupation remains low during this interval. By the end of the Carboniferous, the relative contribution of non-amniote tetrapods is between 50% and 75%, depending on the composition of Amniota used (Fig. 6C; File S5). Following the Gzhelian peak, jaw disparity remains relatively stable until the Roadian (Fig. 6A). More taxa with high PC1 values appear during the Cisuralian, with the diadectid Diadectes lentus as the most extreme example, and the two Bolosaurus species invading more negative PC2 space (Fig. 4B). In addition, this epoch includes the two taxa with the lowest PC1 scores in the sample, the carnivorous Orovenator mayorum and Mesosaurus tenuidens. The Guadalupian, in particular the Capitanian, sees a significant peak in jaw disparity (Fig. 6A). In this time interval, therapsid amniotes start expanding into negative PC2 regions (Fig. 4B), acquiring a bipartite jaw morphology with a well-defined postdentary region. Moreover, the contribution of non-amniote tetrapods is reduced to less than 10%, unless captorhinids are excluded from amniotes (Fig. 6C; File S5). Jaw disparity plummets during the subsequent Lopingian, as morphospace occupation is less dense than in the Guadalupian (Figs. 4B, 6A).

Figure 6 Lateral shape disparity through time, under the baseline phylogenetic hypothesis.

(A) Disparity per time, (B) number of specimens per time bin with the thick line denoting all specimens, fair line amniotes and dashed line non-amniote tetrapods, (C) partial disparity for each major group over time. RFC = Rainforest Collapse, OE = Olson’s Extinction, GE = Guadalupian Extinction. Light goldenrod denotes non-amniote tetrapods, blue denotes synapsids and green denotes sauropsids. Representative silhouettes: Crassigyrinus scoticus for non-amniote tetrapods, Lycosuchus vanderrieti for synapsids and Belebey vegrandis for sauropsids. Specimen sources in File S1.

Occlusal shape morphospace

The first two PC axes describe 64.01% of variation (File S3J). The first nine axes combined describe 95% of the variation. PC1 (42.83%) relates to the curvature of the jaw. Jaws with high PC1 values are flexed laterally. Around PC1 scores of 0, the jaw ramus is straight. In jaws with negative PC1 values, the anterior end of the jaw is flexed medially (File S5). This can occur when the jaw has a broad articular region and a long, wide symphysis. PC2 (21.18%) separates between mediolaterally narrow jaws on the positive axis, to mediolaterally wide jaws on the negative axis.

The jaws of non-amniote tetrapods occupy a greater portion of morphospace than amniotes in occlusal view, regardless of the definition of Amniota used (Fig. 7B; File S3J). In addition, non-amniote tetrapods and amniotes inhabit distinct regions of morphospace considering all four phylogenetic hypotheses (File S3J). Synapsids occupy a greater volume than sauropsids under all definitions, but the centroids are only significantly different when recumbirostrans are included as sauropsids (F = 3.09; p = 0.039). In occlusal view, non-herbivores occupy a greater volume of morphospace than herbivores (W = 333179, p << 0.001), and the centroids are significantly different (F = 5.94; p = 0.001 (Fig. 7F; File S3J)).

Figure 7 PCA and disparity of the occlusal shape analysis, under the baseline phylogenetic hypothesis.

(A, B) Comparing amniotes with non-amniote tetrapods, (C, D) comparing synapsids with sauropsids, (E, F) comparing herbivores with non-herbivores. D. feliceps = Diictodon feliceps; D. magnicornis = Diplocaulus magnicornis; D. paucidens = “Delphaciognathus” paucidens; D. woodi = Doragnathus woodi; E. megacephalus = Eryops mgacephalus; K. kitchingi = Kembawacela kitchingi; M. capensis = Moschops capensis; N. alexanderi = Nochelesaurus alexanderi; P. nazariensis = Procuhy nazariensis; R. gubini = Reiszia gubini.

In PC1, the majority of taxa are found around the midline (Figs. 7A, 7C, 7E), indicating more or less straight jaw rami. Temnospondyls, including Procuhy nazariensis (0.23), Eryops megacephalus (0.34) and Platyrhinops lyelli (0.34), as well as the stem-tetrapods ‘Parrsboro jaw taxon’ (0.21) and Doragnathus woodi (0.48), plot with strong positive values (File S3I). Fewer taxa, all therapsids, are plotted with PC1 values lower than −0.1 (File S3I). These are the dicynodont anomodont Diictodon feliceps (−0.13), the gorgonopsian “Delphaciognathus paucidens” (−0.26) and the tapinocephalid dinocephalian Moschops capensis (−0.52). PC2 sees a more even distribution of taxa. The highest PC2 values, i.e., long and mediolaterally narrow jaw rami, are achieved by various therapsids, like the cynodont Procynosuchus delaharpeae (0.15), the therocephalians Perplexisaurus foveatus (0.15) and Lycideops longiceps (0.16) and the phylogenetically uncertain early therapsid Reiszia gubini (0.17) (File S3I). Conversely, mediolaterally wide jaw rami, corresponding to negative PC2 values, are found in the pareiasaur Nochelesaurus alexanderi (−0.14), the diplocaulid Diplocaulus magnicornis (−0.17) and the dicynodont Kembawacela kitchingi (−0.23) (File S3I).

Temporally, the highest disparity among jaw shapes in occlusal view is observed in the Mississippian (Fig. 8A), despite the broad confidence interval. This is caused by the outlier Doragnathus woodi sitting at the highest PC1 value, as the other four taxa sampled from this interval plot near the midline of PC1 (Fig. 4C). Disparity gradually drops until the Cisuralian (Fig. 8A). During the Pennsylvanian, the diadectid Desmatodon hesperis expands into more negative PC2 values (Fig. 4C), signifying a mediolaterally wide jaw ramus. Strong positive PC1 values, related to strongly curved jaw rami, are exclusive to temnospondyls and the Parrsboro jaw during the Pennsylvanian and Cisuralian epochs. Amniotes remain at low negative PC1 values while expanding into both positive and negative PC2 values during the Cisuralian (Fig. 4C). The relative contribution of non-amniote tetrapods plunges from the Cisuralian onwards (Fig. 8C). A second, smaller, peak in jaw disparity is observed in the Guadalupian (Fig. 8A). The dinocephalian Moschops capensis achieves the lowest PC1 value of our sample during this interval, while other therapsid taxa such as the dicynodont Kembawacela kitchingi and Reiszia gubini reach both extremities of PC2 (Fig. 4C). On the other hand, there are no more taxa with strong positive PC1 values in our sample. By the Lopingian, strong outliers on either side of PC1 have disappeared. The greatest negative outlier is the gorgonopsian “Delphaciognathus paucidens”, while all other taxa are positioned around PC1 values of near 0. Consequently, overall jaw disparity drops into the Lopingian (Fig. 8A).

Figure 8 Occlusal shape disparity through time, under the baseline phylogenetic hypothesis.

(A) Disparity per time, (B) number of specimens per time bin with the thick line denoting all specimens, fair line amniotes and dashed line non-amniote tetrapods, (C) partial disparity for each major group over time. RFC = Rainforest Collapse, OE = Olson’s Extinction, GE = Guadalupian Extinction. Light goldenrod denotes non-amniote tetrapods, blue denotes synapsids and green denotes sauropsids. Representative silhouettes: Crassigyrinus scoticus for non-amniote tetrapods, Lycosuchus vanderrieti for synapsids and Belebey vegrandis for sauropsids. Specimen sources in File S1.

Both our functional and shape analyses show a first increase in jaw disparity during the Moscovian stage (Fig. 3A, 6A). This observation matches the rise of jaw disparity that (Anderson, Friedman & Ruta, 2013) found, starting to increase from the Bashkirian-Moscovian. Notably, the origin of amniotes is dated to the Bashkirian, prior to the Joggins Formation (e.g., Benton & Donoghue, 2007; Mann et al., 2020) and the earliest known herbivorous tetrapod, Desmatodon hesperis, is dated to the Kasimovian epoch (Berman & Sumida, 1995; Reisz & Sues, 2000; www.paleobiodb.org). However, the Pennsylvanian slice of functional morphospace (Fig. 4B) suggests that rather than D. hesperis, carnivorous non-amniote tetrapods, namely the “lepospondyls” Diploceraspis burkei and Oestocephalus amphiuminus and the temnospondyl Adamanterpeton ohioensis, contributed most to the first stage of tetrapod functional jaw diversification. Diploceraspis burkei achieves the highest PC1 value in this time bin while D. burkei, O. amphiuminus and A. ohioensis all expand into positive PC2 values. The amniotes in this interval fall within the same range as the Mississippian non-amniote tetrapods. Similar to the functional morphospace, the lateral shape morphospace sees an expansion into positive PC1 values during the Pennsylvanian (Fig. 4B), caused predominantly by non-amniote tetrapods. The highest PC1 value during this interval is achieved by D. hesperis, and the only two definitive amniotes out of 10 taxa with positive PC1 scores are the synapsids Gordodon kraineri and Sphenacodon ferox. A disparity increase during the Moscovian cannot be replicated with the occlusal morphospace data, as this dataset lacks specimens from the Moscovian-Gzhelian. Instead, it shows a disparity drop between the Mississippian and Pennsylvanian, albeit with a large uncertainty (Fig. 8A).

Relative contribution to overall disparity of non-amniote tetrapods decreases throughout the Pennsylvanian-Permian in all three datasets (Figs. 3A, 6A, 8A). By the end of the Kungurian, relative disparity of non-amniote tetrapods has dropped below 50% in all three datasets under all definitions of Amniota. However, in most analyses, this 50% mark is reached earlier. In the functional disparity analysis, non-amniote tetrapod relative disparity drops below 50% in the Kasimovian when diadectomorphs are considered amniotes. While the decrease of non-amniote tetrapod relative disparity is continuous under most phylogenetic hypotheses, there is a small rebound in the Lopingian when captorhinids, protorothyridids and araeoscelidians are treated as non-amniote tetrapods. This increase is caused by the inclusion of moradisaurine captorhinids, e.g., Moradisaurus grandis and Gansurhinus naobaogouensis.

Discussion

Availability of data

This study uncovers a bias in how the jaws of Palaeozoic tetrapods are figured (File S1). Out of 202 jaws sampled in this study, the vast majority (98.0%) have been figured in lateral view. A little over half (54.9%) have been figured in medial view and only around a third (36.1%) in occlusal view. This is in part attributable to preservational factors: medial and occlusal views are naturally obscured in occluded mandibles. The recent advent of µCT-scanning of occluded Palaeozoic tetrapod skulls (Maddin, Olori & Anderson, 2011; Huttenlocker et al., 2013; Castanhinha et al., 2014; Pardo et al., 2017; Bendel et al., 2018; Gee et al., 2019; Gee, Bevitt & Reisz, 2019, 2021; MacDougall et al., 2019; Rawson et al., 2021; Arbez, Atkins & Maddin, 2022; Hunt et al., 2023; Levy, 2023; Xiong, 2023; Porro, Rayfield & Clack, 2023; Porro, Martin-Silverstone & Rayfield, 2024; Rowe, Bevitt & Reisz, 2023; Jenkins et al., 2024; Ponstein, MacDougall & Fröbisch, 2024) has greatly increased the number of jaw specimens for which medial and occlusal views are available. Nonetheless, published figures of occlusal views are still largely exclusive to specific clades, such as Diadectomorpha, Anomodontia, Captorhinidae and Pareiasauria. Moreover, several descriptive articles lack occlusal and/or ventral views of isolated jaws altogether. Since lateral, medial and occlusal views are applicable to different types of analyses, we urge descriptive articles to consider figuring as many views as possible. We recognise that the inherent anatomical properties of the mandible in some clades problematises figuring certain views: for example, the dentary (and splenial) is a fused element in dicynodont therapsids and a straight-on medial view encompassing the entire surface of the jaw is really only possible in sectioned specimens. With this said, increased use of 3D imaging makes digital sectioning relatively easy, and we recommend this approach in illustrating troublesome taxa such as dicynodonts.

Jaw disparity in amniotes vs. non-amniote tetrapods

Amniotes have greater functional disparity and shape variation in lateral view than non-amniote tetrapods during the Palaeozoic (Figs. 2A, 2B, 5A, 5B), regardless of which of the four compositions of Amniota is used (File S3D, S3G). Here we discuss two hypothesised adaptations specific to amniotes that can help explain this increased disparity.

Within the amniote stem-group, a shift occurs in jaw adductor musculature (e.g., Carroll, 1969; Janis & Keller, 2001). The amniote arrangement allows for more precise biting, including the possibility of tooth-on-tooth occlusion, and the ability to exert pressure while the mouth is closed. We expect this newfound method of mastication to favour extension of the tooth row posteriorly (i.e., to take functional advantage of static-pressure bite), which would increase posterior mechanical advantage. This amniote-type jaw adductor musculature is inferred to have already been fully developed in Diadectomorpha and Protorothyrididae (Carroll, 1969; Janis & Keller, 2001). Even if either clade is placed outside of Amniota, this suggests that the shift in muscle architecture has occurred prior to their divergence with Amniota. Thus, hypothesis 2 (grouping Diadectomorpha and Protorothyrididae with Amniota) is most suitable when comparing changes over the jaw adductor musculature shift. However, we find no significant difference in posterior mechanical advantage between amniotes and non-amniote tetrapods under this definition of Amniota (File S3A). Conversely, under the traditional hypothesis, and when captorhinids and protothyridids are excluded from amniotes, amniotes have a significantly lower posterior mechanical advantage than non-amniote tetrapods. When derived therapsids, with a more posteriorly positioned insertion site, are excluded, the difference between amniotes and non-amniote tetrapods is non-significant (p = 1 for all hypotheses of amniotes). Inclusion of anteriorly positioned neomorphic muscle attachment sites of derived therapsids (e.g., the masseteric fossa) vastly increases both the anterior and posterior mechanical advantage of groups like dicynodonts, therocephalians and gorgonopsians (Singh et al., 2021, 2024), so that the difference with non-amniotes will likely be significant. We found that amniotes have significantly higher anterior mechanical advantage than non-amniote tetrapods when diadectomorphs are included as amniotes (p = 0.037), but not when they are excluded (p = 0.18; File S3A). The increase in anterior mechanical advantage suggests that tetrapods with an amniote-type adductor musculature have relatively shorter pre-adductor fossa jaw lengths than other tetrapods. This is supported by the fact that tooth row length is reduced in Amniota + Diadectomorpha with respect to other tetrapods (p << 0.01; File S3A).

Similarly, amniotes and diadectomorphs have developed costal respiration as an alternative mode of breathing to buccal pumping (e.g., Brainerd, Ditelberg & Bramble, 1993; Brainerd, 1999, 2016; Janis & Keller, 2001; Brainerd & Owerkowicz, 2006). Janis & Keller (2001) hypothesised that buccal pumping imposes constraints on skull morphology, favouring skulls that are optimised in generating negative pressure. Hence, adopting costal respiration is expected to allow amniotes and diadectomorphs to diversify skull morphology in the following ways: (1) adopting dorsoventrally deeper jaw rami, (2) longer symphyses and (3) mediolaterally narrower jaw arches. We found evidence of both (1) and (2) in our results, suggesting that removing the constraints on craniomandibular shape imposed by buccal pumping allowed amniotes and diadectomorphs to evolve more disparate jaw shapes. The clade composed of Amniota and Diadectomorpha bears significantly deeper jaws than other tetrapods, and this is reflected by both the functional and the lateral shape analysis (Files S3D, S3G). In the functional analysis, both average and maximum aspect ratios were found to be significantly higher in amniotes than non-amniote tetrapods, regardless of the definition of amniotes used (File S3G). Amniotes and diadectomorphs extend into higher PC1 values in the lateral shape analysis (Fig. 5A), which is strongly correlated with jaw depth. Furthermore, amniotes have a significantly higher relative symphysial length than non-amniote tetrapods, regardless of which composition of Amniota is used (File 3A). This is in part due to the inclusion of dicynodont anomodonts, which are strong positive outliers on account of their extraordinary long and tightly fused symphyses (Fig. 1I). Even when anomodonts are excluded, amniotes consistently have longer symphyses than non-amniote tetrapods (p < 0.095 for all).

We were unable to find evidence that is either in favour of or against (3) that amniotes have mediolaterally narrower jaw arches than non-amniote tetrapods; jaw arch morphology could not be tested with the Fourier analysis in occlusal view as intended, since silhouettes of complete jaws could not be accurately read in by the Momocs package. We opted to separate jaws at the symphysis, and sample individual jaw rami instead. Consequently, the observed variation pertained mostly to the width and curvature of the jaw ramus, but not the width of the entire jaw arch. Both amniotes and non-amniote tetrapods contain taxa with straight rami and taxa with curved rami (translating to V- and U-shaped jaws, respectively). The increased disparity in non-amniote tetrapods is caused by a few strongly laterally flexed jaws, like in the stem-tetrapod Doragnathus woodi and several temnospondyls (Figs. 7A, 7B). On the other hand, amniotes exhibit greater variation in width of individual jaw rami than non-amniote tetrapods under all definitions of Amniota (Fig. 7A). Accessory dentary shelves like those in moradisaurine captorhinids were not captured by the first two axes of morphospace.

Interestingly, the jaws of synapsids are biomechanically more diverse than those of sauropsids throughout the late Palaeozoic, under each of the four phylogenetic hypotheses (File S3D). A possible explanation is that sauropsids evolved greater variation in the temporal skull region instead (Brocklehurst, Ford & Benson, 2022), allowing more variation in muscle origins and thereby jaw function. In addition, Palaeozoic synapsid jaws are more disparate than sauropsid jaws when viewed in occlusal view (Fig. 7D; File S3J). The sampled sauropsid jaws all have straight rami and are only moderately mediolaterally expanded, resulting in a limited morphospace occupation. Synapsids, on the other hand, inhabit strong negative PC1 values by virtue of the medially flexed jaws of Moschops capensis and “Delphaciognathus paucidens”. In lateral view, sauropsid jaws are more disparate than synapsid jaws unless recumbirostrans are included (Fig. 5D; File S3G). This is not directly apparent from PC1 and PC2, and likely pertains to minor differences in higher PC axes.

Tetrapod jaw disparity throughout the late Palaeozoic

Functional and shape disparity through time show similar overall trends (Figs. 3A, 6A). These trends are explained by major faunal transitions during the Carboniferous and Permian periods. Disparity starts to increase markedly from the Moscovian, similar to the (Anderson, Friedman & Ruta, 2013) disparity curve. The early Permian Gzhelian-Sakmarian interval shows a first major disparity peak in the biomechanical analysis, signalling an episode of tetrapod diversification following the Carboniferous Rainforest Collapse. In the lateral dataset, diversity peaks are reached in the Gzhelian and Sakmarian. By this point, amniotes had already adopted a variety of dietary niches, including herbivory (Sahney, Benton & Falcon-Lang, 2010). In both the biomechanics and the lateral dataset, this corresponds to the development of deeper jaw rami signified by high PC1 values. Disparity gradually declines during the subsequent Artinskian-Kungurian into Olson’s “Gap”. This event marks a major faunal turnover from the Cisuralian faunas dominated by non-amniote tetrapods and “pelycosaurs” to the therapsid and “parareptile”-dominated faunas of the Guadalupian (e.g., Lucas, 2004; Brocklehurst, 2020). Both therapsids and “parareptiles” (specifically pareiasaurs) expanded the functional and lateral shape morphospace. Therapsids maintained deep jaws, while reduction of the adductor muscle insertion site caused an invasion into high PC2 values in the functional morphospace. Regarding the lateral shape morphospace, the complex bipartite therapsid jaws and deep pareiasaur jaws bearing a ventral angular boss increased occupation of shape morphospace into strong negative PC2 space. Both these groups led to a significant peak of both disparities in the Capitanian. This Guadalupian peak in disparity is also observed in the occlusal dataset (Fig. 8A), caused by the flexed jaw of Moschops capensis at the extremity of negative PC1 and a higher variability in mediolateral width of the jaw ramus, expanding into both positive and negative PC2 values. Towards the end of the Capitanian, two pulses of extinctions occurred that wiped out 74–80% of tetrapod genera (e.g., Day et al., 2015; Day & Rubidge, 2021). This extinction notably affected the ecologically diverse therapsid clade Dinocephalia and subclades within Therocephalia and Pareiasauria (Day & Rubidge, 2021). Both functional and lateral shape disparity drop rapidly into the Wuchiapingian and Changhsingian. Herbivorous dinocephalians, such as Moschognathus whaitsi, and basal anomodonts such as Ulemica invisa inhabited positive PC1 yet negative PC2 values during the Guadalupian, with a well-developed adductor fossa and strong force transmission, but this quadrant is vacant during the Lopingian. While dicynodont anomodonts diversified into the Lopingian, this group exclusively inhabited positive PC1 values due to the reduced insertion site of the adductor musculature. Our Lopingian sample accounts for a less dense occupation of morphospace, both in terms of functional and lateral shape, than the Guadalupian sample. This likely reflects a selective extinction (e.g., Puttick, Guillerme & Wills, 2020; Cole & Hopkins, 2021; Polly, 2023) at the end of the Capitanian. Jaw disparity did not recover from the Guadalupian extinctions prior to the end-Permian extinction.

The drop in functional disparity in the Kasimovian is notable (Fig. 3A). Anderson, Friedman & Ruta (2013) did not find this drop in functional disparity, as they used the Kasimovian-Gzhelian as a single time bin. The Kasimovian is the first time bin that is dominated by amniotes (three or four out of five taxa sampled); the diadectid Desmatodon hesperis, the “pelycosaurs” Haptodus garnettensis and Ianthodon schultzei and the araeoscelid diapsid Petrolacosaurus kansensis. Apart from D. hesperis, these early amniotes had remarkably undifferentiated mandibles. The preceding Moscovian included the non-amniote tetrapod outliers Adamanterpeton ohioensis and Oestocephalus amphiuminus, explaining the Kasimovian drop in functional disparity. Yet, functional disparity in the Kasimovian is still greater than in the amniote-lacking time bins (Viséan-Bashkirian). No drop in disparity is observed in the lateral shape disparity; disparity increased from the Moscovian into the Kasimovian.

To ensure our disparity trends are not simply the result of a sampling bias, we analysed the correlation between sample size and disparity. For most of the sampled time span, sample size is decoupled from disparity. Sample size is biased towards certain time bins—the Asselian, Artinskian-Kungurian and Capitanian are particularly well sampled (Figs. 3B, 6B). From the Artinskian, our sample includes mostly amniotes. Our diversity curves (Figs. 3B, 6B, 8B) match published global alpha tetrapod diversity curves (e.g., Ruta & Benton, 2008; Benson & Upchurch, 2013; Dunne et al., 2018; Brocklehurst, 2021). Number of specimens sampled per time bin is relatively stable until the Gzhelian, although both functional and lateral shape disparity start rising from the Moscovian (Figs. 3A, 6A). Functional disparity sees a peak in the Sakmarian but a drop in number of sampled specimens (Figs. 6A, 6B). While functional disparity drops in the subsequent Artinskian, the number of specimens increases. The Kungurian is a particularly well sampled time bin, but both disparity plots show a decrease from the Artinskian. There is a stronger coupling between sample size and disparity in the mid-late Permian. Both the diversity and the disparity plots display a major peak in the Capitanian, followed by a stark drop in the subsequent Wuchiapingian and Changhsingian (Figs. 3A, 3B, 6A, 6B).

Did herbivory relax jaw constraints?

Berks et al. (2025) recently interpreted that the shift to an amniote-type muscle arrangement, as laid out by Janis & Keller (2001), allowed amniotes and diadectomorphs to develop a static pressure bite and facilitate an herbivorous diet. This newfound diet, in turn, would then have relaxed constraints on jaw function and led to the observed increase in jaw disparity in the Late Carboniferous (Berks et al., 2025). Their argument derives from a Pareto optimisation method, in which Berks et al. (2025) found that the jaws of herbivorous tetrapods are “quite optimal” but not optimised under three functional models, and conclude herbivores are less constrained than faunivorous amniotes. This interpretation of Berks et al. (2025) reverses cause and effect relative to our interpretation: that relieving constraints on jaw morphology allowed tetrapods to evolve more diverse jaw shapes and subsequently the ability to explore new dietary niches, including herbivory, successfully.

If the evolution of herbivory in tetrapods was the primary factor that relaxed constraints on jaw shape and caused jaw disparity to increase, we would expect that only the disparity of herbivorous amniotes should increase, and that the disparity of non-herbivorous amniotes would remain similar to that of non-amniote tetrapods. However, we observe that the disparity of non-herbivorous, non-amniote tetrapods experienced a peak between the Moscovian and Sakmarian (File S5), dropping in the Artinskian. In addition, several carnivorous amniotes plot outside the non-amniote morphospace. The first example is the carnivorous sphenacodontid Dimetrodon, which represents an outlier in the jaw biomechanics morphospace relative to the non-amniote and non-diadectomorph tetrapods in the Cisuralian (Figs. 2, 4). Later radiations of non-herbivorous amniotes, such as therocephalian and gorgonopsian therapsids, deviate significantly from the non-amniote tetrapod morphospace, both in the functional and shape analysis (Figs. 2, 5, 7; pbiomechanics = 0.003, plateral = 0.006, pocclusal = 0.006). This is in part due to their neomorphic muscle arrangement (e.g., Kemp, 2009) and bipartite jaw morphology. Moreover, at least within Synapsida, faunivory is the ancestral diet of major evolutionary radiations (Hellert et al., 2023).

Using a greater number of time bins, Berks et al. (2025) recovered a first functional jaw disparity peak in the Gzhelian. After the removal of herbivorous taxa, there is no longer a disparity peak in the Gzhelian, providing evidence for their stated link between herbivory and jaw disparity (Berks et al., 2025). By contrast, in our functional and lateral shape analysis, we find a first initial increase in the Moscovian (Figs. 3A, 6A), which is correlated to an expansion in morphospace occupation by faunivorous non-amniote tetrapods (Figs. 4A, 4B). Secondly, Berks et al. (2025) found that only herbivorous amniotes increase in disparity during the Capitanian, and thus attribute the Capitanian peak to their diversification. While the Capitanian peak is also prominent in our analysis (Figs. 3A, 6A), we find that faunivorous therapsids in particular expand in large regions of morphospace (Figs. 4A, 4B). The therapsid expansion into high PC2 values in our biomechanical morphospace reflects low jaw closing mechanical advantage, which is an artefact of our measurement of the adductor musculature. In the lateral shape morphospace, we find a strong separation between therapsids and other tetrapods along PC2, resulting from the bi-partite jaw morphology in therapsids. This differs from Berks et al.’s (2025) results, as their PC2 distinguishes between dorsally and ventrally flexed jaws.

Additionally, we note some issues with the assumptions and methodological decisions Berks et al. (2025) made in preparing their Pareto optimisation. First, the Berks et al. (2025) study is limited to jaws in lateral view, which does not take into account the three-dimensional aspect of tetrapod jaws. In preparing the Pareto optimisation, Berks et al. (2025) assumed that aquatic taxa minimise skull depth to reduce hydrodynamic drag, which, they hypothesise, also affects jaw depth. However, aquatic habitats do not necessarily force dorsoventrally shallow jaws, even for nektonic taxa. For example, fishes adapted a wide range of jaw depths throughout their evolutionary history (Hill et al., 2018). Palaeozoic fish groups with high bite-force, such as various “placoderms”, developed deeper jaws that are more resistant to dorsoventral forces (Anderson, 2009). Similarly, diverse clades of secondarily aquatic amniotes bear a tall coronoid process on the jaw, such as trionychid turtles (e.g., Evers et al., 2023; Ponstein et al., 2024), placodont sauropterygians (e.g., Diedrich, 2013; Neenan et al., 2015) and the rhynchocephalian Ankylosphenodon pachyostosus (Reynoso, 2000). Thus, a dorsoventrally shallow mandible is not enforced in nektonic aquatic taxa—a deeper mandible need not compromise skull height or hydrodynamic performance, and other biological factors may favour deepening of the mandible. Conversely, Berks et al. (2025) removed jaw depth as a factor in the terrestrial optimisation models, assuming terrestrial taxa have no preference to either dorsoventrally shallow or deep jaws. While aerodynamic forces might indeed have negligible effect on skull and jaw height, jaw robustness is correlated strongly with dietary specialisation in various terrestrial amniote clades (e.g., Stubbs et al., 2013; Button & Zanno, 2020; Singh et al., 2021) and thus should be an important factor in modelling jaw optimisation.

Finally, Berks et al. (2025) noted that there might have been alternative factors not captured by their model that played a role in lifting constraints on jaw shape, such as a fully terrestrial feeding system, a remodelled hyobranchial apparatus morphology, and a long, mobile neck. Both the earlier mentioned amniotic jaw adductor musculature and a mobile neck are linked to costal respiration (Janis & Keller, 2001). Janis & Keller (2001) argued that an amniotic muscle arrangement, whereby the adductor muscle mass is differentiated into a pterygoideus and posterior adductor complex, would not function properly in a flat-headed non-amniote tetrapod. Operating on a dorsoventrally flat cranium, the posterior adductor complex muscles would simply be too short to depress the lower jaw. The rearrangement in jaw musculature could not have been completed until dorsoventrally deeper crania evolved, removed from the constraints imposed by buccal pumping (Janis & Keller, 2001). Secondly, a longer neck consisting of more than five cervical vertebrae, which is absent in non-amniote and non-diadectomorph tetrapods, necessitates a trachea-like structure that in turn favours a more efficient mode of breathing like costal respiration (Janis & Keller, 2001). The connection between the shift in primary mode of breathing and the lifting on constraints on jaw morphology suggests that the origins of costal respiration, and not herbivory, permitted the observed expansion in Palaeozoic tetrapod jaw disparity.

Conclusions

Using a greatly expanded dataset, we were able to replicate the Pennsylvanian rise in functional jaw disparity that (Anderson, Friedman & Ruta, 2013) observed, and continue the trend into the Permian. The earliest diversification in biomechanical disparity is during the Moscovian, and is caused by carnivorous non-amniote tetrapods. From the early Permian onwards, amniote jaws are more disparate than non-amniotes and the relative contribution of non-amniotes to overall disparity declines. Overall, functional and lateral shape disparity of tetrapod jaws follow similar patterns throughout the Palaeozoic, which are explained by major faunal turnovers and extinction events. Amniotes have deeper jaw rami and a shorter pre-coronoid jaw length than non-amniote tetrapods—leading to an increased anterior mechanical advantage. This likely stems from a restructuring of adductor jaw musculature in amniotes, and releasing a constraint on skull shape imposed by buccal pumping. Herbivores in turn have higher mechanical advantage and jaw depth than non-herbivores, reaching values that are not achieved by any non-herbivorous tetrapod. Thus, our analysis demonstrates that it was the radiation of amniotes and the freeing of constraints acting on craniomandibular shape that facilitated their ecomorphological diversification, enabling terrestrial tetrapods to exploit crucial new dietary niches as herbivory and promoting the flowering of tetrapod life on land in the late Palaeozoic.

Supplemental Information

Supplemental Information 1 List of specimens.

Supplemental Information 2 Description of measurements.

Supplemental Information 3 Stats.

Supplemental Information 4 Raw measurements.

Supplemental Information 5 Specimen sources.

Supplemental Information 6 Rscripts required for the biomechanical, lateral shape and occlusal shape analyses - including necessary text files and jaw silhouettes.

Supplemental Information 7 Boxplot.

Comparing biomechanical traits between amniotes (darkgoldenrod) and non-amniote (lightgoldenrod) tetrapods under phylogenetic hypothesis 2 (H2; Diadectomorpha are synapsid amniotes), hypothesis 3 (H3; Captorhinidae and Araeoscelidia are non-amniote tetrapods) and hypothesis 4 (H4; Recumbirostra are sauropsid amniotes).

Supplemental Information 8 Boxplots.

Comparing biomechanical traits between non-amniote tetrapods (lightgoldenrod), sauropsids (green) and synapsids (blue) under phylogenetic hypothesis 2 (H2; Diadectomorpha are synapsid amniotes), hypothesis 3 (H3; Captorhinidae and Araeoscelidia are non-amniote tetrapods) and hypothesis 4 (H4; Recumbirostra are sauropsid amniotes).

Supplemental Information 9 PCA PC1-2 of jaw biomechanics under all four phylogenetic hypotheses.

A = baseline hypothesis; B = hypothesis 2 (Diadectomorpha are synapsid amniotes); C = hypothesis 3 (Captorhinidae and Araeoscelidia are non-amniote tetrapods); D = hypothesis 4 (Recumbirostra are sauropsid amniotes).

Supplemental Information 10 PCA PC1-3 of jaw biomechanics under all four phylogenetic hypotheses.

A = baseline hypothesis; B = hypothesis 2 (Diadectomorpha are synapsid amniotes); C = hypothesis 3 (Captorhinidae and Araeoscelidia are non-amniote tetrapods); D = hypothesis 4 (Recumbirostra are sauropsid amniotes).

Supplemental Information 11 Disparity of jaw biomechanics under all four phylogenetic hypotheses.

A = baseline hypothesis; B = hypothesis 2 (Diadectomorpha are synapsid amniotes); C = hypothesis 3 (Captorhinidae and Araeoscelidia are non-amniote tetrapods); D = hypothesis 4 (Recumbirostra are sauropsid amniotes).

Supplemental Information 12 PCA PC1-2 of jaw biomechanics under all four phylogenetic hypotheses.

A = baseline hypothesis; B = hypothesis 2 (Diadectomorpha are synapsid amniotes); C = hypothesis 3 (Captorhinidae and Araeoscelidia are non-amniote tetrapods); D = hypothesis 4 (Recumbirostra are sauropsid amniotes).

Supplemental Information 13 Disparity of jaw biomechanics under all four phylogenetic hypotheses.

A = baseline hypothesis; B = hypothesis 2 (Diadectomorpha are synapsid amniotes); C = hypothesis 3 (Captorhinidae and Araeoscelidia are non-amniote tetrapods); D = hypothesis 4 (Recumbirostra are sauropsid amniotes).

Supplemental Information 14 Explanation of first two PC axes.

A lateral shape morphospace and B occlusal shape morphospace

Supplemental Information 15 PCA PC1-2 of the lateral shape analysis under all four phylogenetic hypotheses.

A = baseline hypothesis; B = hypothesis 2 (Diadectomorpha are synapsid amniotes); C = hypothesis 3 (Captorhinidae and Araeoscelidia are non-amniote tetrapods); D = hypothesis 4 (Recumbirostra are sauropsid amniotes).

Supplemental Information 16 PCA PC1-3 of the lateral shape analysis under all four phylogenetic hypotheses.

A = baseline hypothesis; B = hypothesis 2 (Diadectomorpha are synapsid amniotes); C = hypothesis 3 (Captorhinidae and Araeoscelidia are non-amniote tetrapods); D = hypothesis 4 (Recumbirostra are sauropsid amniotes).

Supplemental Information 17 Disparity of the lateral shape analysis under all four phylogenetic hypotheses.

A = baseline hypothesis; B = hypothesis 2 (Diadectomorpha are synapsid amniotes); C = hypothesis 3 (Captorhinidae and Araeoscelidia are non-amniote tetrapods); D = hypothesis 4 (Recumbirostra are sauropsid amniotes).

Supplemental Information 18 PCA PC1-2 of the lateral shape analysis under all four phylogenetic hypotheses.

A = baseline hypothesis; B = hypothesis 2 (Diadectomorpha are synapsid amniotes); C = hypothesis 3 (Captorhinidae and Araeoscelidia are non-amniote tetrapods); D = hypothesis 4 (Recumbirostra are sauropsid amniotes).

Supplemental Information 19 Disparity of the lateral shape analysis under all four phylogenetic hypotheses.

A = baseline hypothesis; B = hypothesis 2 (Diadectomorpha are synapsid amniotes); C = hypothesis 3 (Captorhinidae and Araeoscelidia are non-amniote tetrapods); D = hypothesis 4 (Recumbirostra are sauropsid amniotes).

Supplemental Information 20 PCA PC1-2 of the occlusal shape analysis under all four phylogenetic hypotheses.

A = baseline hypothesis; B = hypothesis 2 (Diadectomorpha are synapsid amniotes); C = hypothesis 3 (Captorhinidae and Araeoscelidia are non-amniote tetrapods); D = hypothesis 4 (Recumbirostra are sauropsid amniotes).

Supplemental Information 21 Disparity of the occlusal shape analysis under all four phylogenetic hypotheses.

A = baseline hypothesis; B = hypothesis 2 (Diadectomorpha are synapsid amniotes); C = hypothesis 3 (Captorhinidae and Araeoscelidia are non-amniote tetrapods); D = hypothesis 4 (Recumbirostra are sauropsid amniotes).

Supplemental Information 22 PCA PC1-2 of the occlusal shape analysis under all four phylogenetic hypotheses.

A = baseline hypothesis; B = hypothesis 2 (Diadectomorpha are synapsid amniotes); C = hypothesis 3 (Captorhinidae and Araeoscelidia are non-amniote tetrapods); D = hypothesis 4 (Recumbirostra are sauropsid amniotes).

Supplemental Information 23 Disparity of the occlusal shape analysis under all four phylogenetic hypotheses.

A = baseline hypothesis; B = hypothesis 2 (Diadectomorpha are synapsid amniotes); C = hypothesis 3 (Captorhinidae and Araeoscelidia are non-amniote tetrapods); D = hypothesis 4 (Recumbirostra are sauropsid amniotes).

Supplemental Information 24 Functional disparity.

A amniotes (fair) versus non-amniote tetrapods (dashed line) and B carnivorous tetrapods (dashed line) through time.

Supplemental Information 25 Lateral shape disparity.

A amniotes (fair) versus non-amniote tetrapods (dashed line) and B carnivorous tetrapods (dashed line) through time.

Supplemental Information 26 Partial disparity of tetrapod jaw biomechanics through time, per clade.

A = hypothesis 2 (Diadectomorpha are synapsid amniotes); B = hypothesis 3 (Captorhinidae and Araeoscelidia are non-amniote tetrapods); C = hypothesis 4 (Recumbirostra are sauropsid amniotes). Lightgoldenrod = non-amniote tetrapods, green = Sauropsida, blue = Synapsida.

Supplemental Information 27 Partial disparity of tetrapod jaw lateral shape through time, per clade.

A = hypothesis 2 (Diadectomorpha are synapsid amniotes); B = hypothesis 3 (Captorhinidae and Araeoscelidia are non-amniote tetrapods); C = hypothesis 4 (Recumbirostra are sauropsid amniotes). Lightgoldenrod = non-amniote tetrapods, green = Sauropsida, blue = Synapsida.

Supplemental Information 28 Partial disparity of tetrapod jaw occlusal shape through time, per clade.

A = hypothesis 2 (Diadectomorpha are synapsid amniotes); B = hypothesis 3 (Captorhinidae and Araeoscelidia are non-amniote tetrapods); C = hypothesis 4 (Recumbirostra are sauropsid amniotes). Lightgoldenrod = non-amniote tetrapods, green = Sauropsida, blue = Synapsida.

Supplemental Information 29 Partial disparity.

carnivores (firebrick) versus herbivores (darkolivegreen) in terms of A jaw biomechanics and B lateral shape.

We thank T. Hübner for access to the collection of the Friedenstein Stiftung Gotha (MNG), and S. König for photographing specimens from this collection. F. Söderblom is thanked for helpful discussions, and J. Renaudie for assisting with R. Lastly, we are grateful to S. Singh and an anonymous reviewer for their thorough review that greatly benefited this manuscript.

Additional Information and Declarations

Competing Interests

The authors declare that they have no competing interests.

Author Contributions

Jasper Ponstein conceived and designed the experiments, performed the experiments, analyzed the data, prepared figures and/or tables, authored or reviewed drafts of the article, and approved the final draft.

Mark J. MacDougall conceived and designed the experiments, analyzed the data, authored or reviewed drafts of the article, and approved the final draft.

Joep Schaeffer analyzed the data, authored or reviewed drafts of the article, and approved the final draft.

Christian F. Kammerer analyzed the data, authored or reviewed drafts of the article, and approved the final draft.

Jörg Fröbisch conceived and designed the experiments, analyzed the data, authored or reviewed drafts of the article, and approved the final draft.

Data Availability

The following information was supplied regarding data availability:

The code is available in the Supplemental File.

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
