# Peer review of "Mandibular form and function is more disparate in amniotes than in non-amniote tetrapods from the late Palaeozoic"

_PeerJ, doi:10.7717/peerj.20243_

## Round 0.1 · original submission · Minor Revisions

· Academic Editor

Minor Revisions

After careful evaluation of your submission and considering the reviewers’ comments, we are pleased to inform you that your manuscript has been recommended for publication pending minor revisions. Both reviewers agree that the study is relevant and novel, and that it makes a valuable contribution to the field.

However, we would like to draw your special attention to the comments made by Reviewer 2, who suggests a series of minor but meaningful revisions aimed at improving the clarity, structure, and scientific impact of the manuscript. In particular, please pay close attention to the suggestions regarding the reorganization and rewriting of sections of the main text, Results and Discussion, as well as the figures. Additionally, the recommendations made in the Materials and Methods section should be carefully addressed.

I kindly ask you to revise your manuscript accordingly and provide a detailed point-by-point response to the reviewers’ comments with your resubmission.

Reviewer 1 ·

Basic reporting

Overall very well done. I just have a couple comments concerning the writing:

Lines 166-173: Even though full definitions of the traits measured are included in the supplemental materials, the nine traits should still be listed here in the methods. At no point in the main manuscript are the 9 traits actually defined/listed even in a shorten form.

Lines 290-335: This is a large paragraph that appears to simply list values and ranges for the various traits in specific taxa. Given that the manuscript is a bit long already, i would cut this paragraph entirely or focus only on specific values that are unusual or support the main hypothesis. If the authors want to keep the information, I feel that it could be made into a table that lists max and min for each trait with the taxon in question included.

Lines 428-447: This is the same as above, the paragraph seems unnecessary.

Experimental design

Overall very solid. My only concern is that the authors verify that the classification criteria used to define herbivores does not utilize the same morphological measurements that the 9 functional traits are based on.

Validity of the findings

No comment

Additional comments

Typo on lines 274-275: It states: "Under this hypothesis, amniote variance is higher in all traits but tooth row length (Figure 1; Supplementary File 3A)." In figure 1, the variance is not being compared, just the values.

·

Basic reporting

This manuscript presents valuable findings and rather detailed analyses with extensive reference to published literature, particularly when providing the rationale for the investigation in the introduction. The figures are professional but a little rudimentary and would benefit from some reorganisation to both reduce the overall number of figures and make them more intuitive for readers. The authors have beautifully put together their data in the supplementary materials and should be congratulated on their organisation.
The writing would strongly benefit from revision with a focus on clarity, conciseness, and connection to the core research questions (as outlined in the introduction) to improve the structure, flow and effectiveness of the manuscript. There’s a tendency throughout the manuscript to use many short sentences in quick succession, which can create a somewhat monotonous writing style. This is especially apparent in the first paragraph of the introduction, which needs to really hook the reader. I believe that its overall readability and impact can be greatly improved through more effective articulation of the results and their broader implications.

Experimental design

This manuscript provides a comprehensive analysis of jaw shape and function in terrestrial tetrapods through the late Palaeozoic. The extensive taxon sampling, breadth of functional measurements, and integration of multiple morphological aspects are commendable, and greatly expand on previous studies. It addresses important evolutionary questions on the drivers of early terrestrial tetrapod diversification, and the role of different intrinsic/biotic constraints, which are clearly outlined in the introduction.
The methods are clear and well detailed, particularly in the supplementary methods. The morphometric methods applied are solid with a strong track record of use in previous studies. I would recommend some additional statistical tests focused on the differences between amniotes and anamiotes through time to better support their arguments on the drivers of tetrapod disparity in the late Carboniferous (see my detailed materials and methods comments). Nonetheless, the present work is clearly the product of significant effort by the authors and packed with important findings and observations for palaeobiology/evolutionary biology researchers. As such, I congratulate them on putting together a very interesting study.

Validity of the findings

The findings presented in this study are numerous and interesting, with reference to the original research question. I think there's a lot of information contained within this study, but the presentation of so many of these findings in the results and discussion dilutes the core findings of the study with respect to the core research questions. I appreciate that reporting on such an taxonomically and temporally expansive study is tricky, but I would recommend rewrites to focus more on the big picture findings, as well as reorganisation and redrafting of combined figures to address this issue. (Please see my detailed comments on the results and discussion sections). The study makes an important contribution to our understanding of the drivers of ecomorphological diversity, with timely relevance to the current debate owing to the contrasting patterns and interpretations with recently published literature. I strongly endorse the publication of this study following some revisions as outlined in my detailed comments.

Additional comments

General Overview: This manuscript provides a valuable, comprehensive analysis of jaw shape and function in terrestrial tetrapods through the late Palaeozoic. The extensive taxon sampling, breadth of functional measurements, and integration of multiple morphological aspects are commendable, and greatly expand on previous studies. It addresses important evolutionary questions on the drivers of early terrestrial tetrapod diversification, and the role of different intrinsic/biotic constraints. It is clearly the product of significant effort by the authors and is packed with important findings and observations for palaeobiology/evolutionary biology researchers. As such, I congratulate them on putting together a very interesting study.
However, to be suitable for publication, I recommend various minor but meaningful revisions to improve clarity, structure, and scientific impact. I think going through and editing the manuscript with a focus on clarity, conciseness, and connection to the core research questions (outlined in the introduction) will greatly improve the structure, flow and effectiveness of the writing, helping to more effectively articulate the results’ implications and significantly enhancing the overall readability and impact of this manuscript. There’s a tendency throughout the manuscript to use many short sentences in quick succession, which can create a somewhat monotonous writing style. This is especially apparent in the first paragraph of the introduction, which needs to really hook the reader. To address this, try to combine and condense sentences to produce smoother prose and improve focus on key findings by cutting redundant or minor details. I also recommend some additional statistical tests focused on the differences between amniotes and anamiotes through time (see materials and methods comments), and reorganisation of the figures (see results and figures comments).
Overall, I believe that this study deserves publication with PeerJ following minor revisions as outlined in my comments below. I also welcome the authors contacting me for any further clarification and wish them the best of luck in getting this study published in good time.

Detailed Comments

Introduction:
I’d recommend rephrasing the opening of the intro to avoid many short sentences in quick succession following my general comment above on writing style. I’d also recommend some reordering of content. I’d combine paragraphs 1 & 2 and cut their content down to get more quickly to the rationale of the present study (currently in paragraph 3.
Line 40-44: The sentences of this opening paragraph are rather short and seem a bit disconnected from each other. Also, they’re short and in quick succession, which creates a monotonous writing style that doesn’t really entice prospective readers – this is the opening and so needs to hook the reader and provide clear context and/or rationale for the study. I’d try rephrasing these sentences and combining this paragraph with the current second paragraph of the intro. I’d also remove “The initial fish-tetrapod transition of the Middle-Late Devonian is an example of an adaptive.” as it doesn’t really connect to the present study.
Line 51-53: “flow relative to the head to capture and swallow food, including behaviours such as filter feeding and suction feeding. These strategies are ineffective on land in the far less dense medium of air, favouring jaw prehension instead.” These sentences are a little unclear and can be improved by rephrasing to something along the lines of: “flow relative to the head to capture and swallow food, enabling trophic behaviours such as filter and suction feeding. However, these feeding strategies are ineffective in air, requiring feeding on land to use jaw prehension.” I would also add some brief prior explanation for why the paper focuses on feeding and the mandible (e.g., it’s a critical biotic function and the lower jaw is functionally dedicated to capturing/processing food).
Line 66: Use “million years or Myr” instead of “Ma” here as talking about a timespan not an age range.
Line 75-76: I’d rephrase this to something along the lines of: “Two major differences in amniote and anamniote jaw function may explain the delay and subsequent acceleration of jaw evolution.”
Line 75-101: Combine these paragraphs into one and cut down to be more concise by rephrasing: e.g., “Firstly, the restructuring of the jaw adductor musculature from anamniotes to amniotes provided amniotes with a static-pressure bite, allowing them to exert pressure when the mouth is closed, and exhibit greater biting control and dental occlusion (Carroll 1969; Janis and Keller 2001; Reisz 2006). These traits facilitated more mechanically efficient jaws, powering greater biting forces, particularly at the back of the jaw.”
I also think that greater explanation/outlining is required for how these two key differences directly led to the pattern of a delay and then acceleration in jaw disparity growth. Why would different jaw muscles and respiratory systems lead to this pattern of morphological diversification? It seems like you’re hinting at the evolution of amniotes as being the key event in terrestrial tetrapod jaw diversification, but it would help the reader if these links were explained a bit more clearly.
Line 121-122: Rephrase “filling in taxonomic gaps and extending the disparity plot into the subsequent Permian period” to “filling in taxonomic gaps and charting disparity patterns into the subsequent Permian period”.

Materials & Methods:
I’d recommend including a taxonomic breakdown of the sampling to show how many taxa were sampled for each of the major clades. I would also commend the authors on their supplementary methods and the very clear outlining of how they took their measurements. Also, given a key objective of this paper is testing the drivers of the delayed rise in terrestrial tetrapod jaw disparity and trying to link it to the evolution of amniotes, I think it would be worthwhile testing for significant differences in amniotes and anamniotes through time/per timebin. At present the comparison doesn’t give a clear idea of how both groups progress across the temporal trend under investigation. Following this thread, I also think it would be worth charting independent disparity curves for amniotes and anamniotes. Furthermore, restricting the occlusal view shape analysis to individual rami somewhat precludes analysis of the ‘v’ or ‘u’ shaped jaw difference between amniotes and amamniotes, and this is acknowledged in text later in the discussion – given that you have the individual rami, could these not be duplicated and arranged, based on broader observation of the relevant taxon material and nearest relatives, to create models of the undeformed jaws in the occlusal view?
Line 135: See my above comment on usage of Ma and Myr.

Results:
There’s a lot of information presented in the results, and much of it is interesting and relevant, but these points are somewhat overwhelmed in places by the high level of detail given; try to focus on the key points and cut unnecessary details (e.g., multiple examples of taxa related to aspects of morphospace variation - let the figures do some of this work by labelling the taxa of interest). This issue is particularly pressing in the paragraphs at lines 290-335 and 358-380. Also try to avoid talking about taxon positions only in terms of PC position – try to also include or talk mainly about their actual morphology/functionality as in lines 375-378. It may help to try defining key regions of morphospace with an intuitive name that can then be used subsequently. I also think the current layout of the results can be amended to improve the succinctness of the results section and clarity of the comparisons between different aspects of jaw morpho-function. I’d strongly recommend attempting to combine the plots for the occlusal and mesial view morphospace and disparity - The boxplots currently taking up half of the figure space don’t need to be that large and can easily be reduced in size, perhaps even becoming slightly offset subpanels within the morphospace plots. This would create clearer distinction between the morpho-function and shape parts of the study, and better outline comparative trends. At present, outlining three separate sets of results that largely correspond is creating much repetition. Plus, this would ensure that two thirds of the critical results can be included in the main text rather than placed in the supplement; I appreciate that not everything can be placed in the main text, but one of the biggest strengths of this study is the analysis of multiple aspects of jaw morpho-function, yet most of these results are somewhat ‘hidden’ in the supplement!
Line 358-380: This paragraph appears to be focusing on the differences between amniotes and anamniotes with specific details of the relevant subclades, but the key message of amniotes having greater disparity is being obscured by presenting the stats of so many different taxa. I’d recommend rephrasing and cutting down this section by trying to focus on the key patterns of variation with respect to the larger/key ecological and taxonomic groups, with the specific details of only a few critical taxa included where necessary to support the main point. Cutting down the length of this section will help improve the clarity of the messaging.
Line 370: Two diadectids and a diplocaulid are mentioned as outliers – I’d add some explanation of what aspect of morphology makes them outliers.
Line 392-393: Rephrase “inhabit positive PC1 scores” as these taxa can have/exhibit positive PC1 scores or inhabit positive PC1 regions/morphospace. Also, I’d explicitly mention overlap between carnivores and herbivores rather than just saying dietary categories as this is much clearer and avoids confusion if readers missed the earlier point about PC1 positive space being dominated by herbivores.
Line 398-399: Why not quantitatively measure the overlap through time of herbivores and non-herbivores using the distance in centroid and/or SoV? This will give greater weight to the argument that herbivory is a key driver of increasing disparity.
Line 428-447: Again, try to avoid talking only about PC values – keep the actual morphology in mind as this helps the reader to understand the patterns of variation independently of the morphospace plot. What do low or high values correspond to in terms of shape and/or clade-wise patterns of morphospace occupation?
Line 461: Typo – insert “the” so it reads “as the most extreme example”.
Line 461-462: “invade more negative PC2 space” not “invade stronger negative PC2 values”.

Discussion:
While there is a need to reiterate and explain results in the discussion, there’s a lot of results description, leading to a blurring of results and discussion at present. Much of these results are important and definitely belong in the paper, but should be in the results section, with the discussion featuring the implications and broader relevance of said results. There’s also a tendency to repeat points from the intro or results, with the relevant implications from the present study being presented much later and without much detailed explanation; it is these implications that should be the main points of the discussion. Therefore, try to be more concise when reiterating points from the intro and results, so that you can more quickly get to the point of what your results mean for those ideas from the previous sections. Also, give more space and detail to what the present study’s results mean for these ideas. Do they support the statements from the intro? Do these results back up claims supporting or refuting previously stated ideas? Guide the reader through your thought process. Most of the details given on results in this section are interesting and relevant but should be placed in the results section (e.g., see line 549-552).
Line 532-537: This is almost directly repeating the latter half of the introduction. Cut this back to provide only a brief summary and rephrase to keep emphasis what your results contribute to this point.
Line 539-540: Change “Even if either clade is placed outside of Amniota, this suggests that the shift in muscle architecture has occurred prior to the divergence between Amniota and said clade.” to “Even if either clade is placed outside of Amniota, this suggests that the shift in muscle architecture has occurred prior to their divergence with Amniota.” to avoid using “said clade”, which is a little archaic.
Line 543-544: The key finding that there is no significant difference in posterior MA between amniotes and anamniotes should be presented earlier in the paragraph instead of buried midway – To get to the point more directly, I’d reframe the current opening to something along the lines of: “The shift jaw adductor musculature from non-amniote to amniote tetrapods (Carroll 1969; Janis and Keller 2001) is heralded as a key structural change that enabled amniote trophic diversification on land and the increase in jaw disparity reported by previous studies (Anderson et al. 2013; Berks et al. 2025). However, we find no significant difference in posterior mechanical advantage between amniotes and non-amniote tetrapods under any definition of Amniota (Supplementary File 3A)”.
Line 548: Avoid referring to “this group” or “said group” and use the actual name of the group for clarity.
Line 549: If you think it’s likely to give a significant difference then why not test it, seeing as you have the data?
Line 537-542/551-556: I think it would be better to try tying these two sections together and lightly rephrasing to focus more on the placement of diadectomorphs within or outside Amniota, and how this affects the idea that the amniote condition supported the radiation of jaw morphologies. If possible, it may be better to try and combine this point with the following paragraph, which also focuses on the traits of diadectomorphs and amniotes.
Line 565-577: Good detail, but what does this mean for tetrapod terrestrialisations and the role of the origin of amniotes in driving recovered disparity trends? Be sure to stress the links back to the key questions under investigation in the paper.
Line 579-604: What do these differences mean in terms of jaw functionality and how does this relate back to patterns of tetrapod/amniote diversification? These questions should be answered, perhaps in a final summary paragraph to finish off the whole amniote vs non-amniote section, giving the reader the key findings and their broader implications.
Line 606-637: These two paragraphs contain a really good level of detail, but this content belongs in the results section.
Line 611: Desmatodon is no longer the earliest known herbivorous tetrapod – the edaphosaur, Melanedaphodon hovaneci dates to the Moscovian (Mann et al., 2023. Scientific Reports). Melanedaphodon also has a fairly well-preserved jaw, and a reconstruction figured in the paper, so may be worthwhile including given its prominence as the earliest known tetrapod herbivore to help bolster your temporal patterns of jaw disparity and parts of your argument pertaining to the interpretations of Berks et al. (2025).
Line 639-671: This paragraph begins to focus more on the drivers of disparity trends by linking to key geological events such as the end-Capitanian mass extinction but again the prose focuses more on describing results than explaining them. Which areas of morphospace are vacated/colonised following ‘X’ event? What do these changes in morphospace occupation/disparity mean for the trophic ecologies of the groups that survived/went extinct? Furthermore, what do the answers to these questions mean for understanding the broader drivers of terrestrial tetrapod diversification?
Line 660-661: Why is Dinocephalia referred to as a clade, but Therocephalia and Pareiasauria are subclades? They’re all clades unless referred to in the context of the overarching clade.
Line 679: Amend to “explaining the Kasimovian drop in functional disparity” as may be misconstrued as meaning the outliers caused a drop in disparity in the Moscovian.
Line 685-698: Please provide more clarification/detail on why patterns of correspondence between sampling and disparity is relevant to the aims of the present paper – at present, this section is presented without context.
Line 700-772: This section on “Did herbivory relax jaw constraints?” is exactly the sort of content and writing style that should be present throughout the entire discussion section. It’s direct and refers to both wider literature and the present study to advance a distinct argument that directly relates back to the core questions of the study.
Line 712-721: Be careful when making comparisons between amniotes and anamniotes as the comparisons are supposed to be in the context of the increase in disparity in the later Carboniferous, but you refer to therapsids, which largely date to the mid-late Permian. Look to your temporal patterns of morphospace occupation and disparity trends (through the Carboniferous) to support your argument here. This is also why more statistical assessment of the differences between key ecological groups/clades for each timebin would be beneficial to the present study.
Line 723-732: Good presentation of the contrasting findings between the present study and Berks et al. 2025. I’d add in some explanation of why the results differ, particularly in respect to the composition of the Capitanian disparity peak.
Line 734-754: I think you’ve highlighted a critical issue on the assumptions made by Berks et al. (2025), and I think you need to bring in your results here as well as the examples you’ve given from the literature to stress the point that their assumptions on what makes an optimal aquatic jaw is too simplistic.
Line 769-772: This is a good point, but move it to earlier in the paragraph, with the relevant explanation following afterwards rather than beforehand so that the reader is clear on the context of the details given. Also, refer to your findings here to show that your argument is backed by hard data.
Line 774-794: While this paragraph is noteworthy from a practical point of view, I think it would be better placed in the material and methods alongside the rationale for image and functional measurement sampling. The closing part of the discussion should leave the reader with the most critical takeaway points, and the end of the previous section was a more powerful point to end on.

Conclusions:
Overall, it’s quite good but I think the language could be a bit sharper to really hammer home the importance of the study’s findings, and leave the reader with clear takeaway points.
Line 804: the role of major faunal turnovers and extinction events are not really discussed in the discussion beyond description of disparity trends/morphospace occupation patterns aligning with these events. This should be remedied – see my comments relating to Line 639-671.
Line 809-812: The final sign-off sentence is currently a little vague and a bit ineffectual – this is the final message to the reader so make it clear, direct and give it some swagger! Rephrase to something like: “Thus, our analysis demonstrates that it was the diversification of amniotes and the evolution of their innovative jaw musculature that facilitated their ecomorphological diversification, enabling terrestrial tetrapods to exploit new dietary niches and promoting the flowering of tetrapod life on land in the late Palaeozoic.”

Acknowledgements:
Don’t forget to mention any relevant funding/grants that supported the study.

Figures:
I think that many of these figures can be combined by reducing certain elements (particularly the boxplots) down in size. Fewer figures with more information packed into them would be easier for readers to navigate while reading. Given there’s a much repetition of the figures for each of the different aspects of jaw morpho-function, my comments for Figs. 1-4 should be considered as appropriate for all the versions of the relevant figure.
Figures 1 & 2: I’d combine these two figures - These boxplots can be greatly reduced in size enabling the boxplots to be combined across each functional character, so that you have all four groups alongside each other. This is more spatially efficient and is easier for readers to read and compare across the different groups.
Figure 3: Perhaps include convex hulls to help highlight the differences in morphospace occupation. Also, again the boxplots are rather large and should be reduced in size to give more space for the morphospaces, which are the focal points of the figure. Making the morphospaces larger would also allow further annotations if wanted.
Figure 4: I’d subsume panel C into a subpanel of panel A and then add a new panel showing the sum of variance of the different major groups (amniotes and anamniotes) enabling description of the key differences in disparity between these groups alongside the temporal patterns of morphospace occupation shown in Figure 5. This will make for more coherent and detail comparison between disparity curve and morphospace occupation trends. Also, I don’t think the information in panel B really warrants a whole panel – why not include this info as mini pie charts for each timebin within Panel A to be more efficient with your figure space? I’d also include the key geological events such as mass extinctions.
Figure 5: Again, I’d recommend including convex hulls to help highlight the differences in morphospace occupation through time. Also, add in the years alongside the timescale, with the key geological events highlighted as suggested for Fig. 4.

Supplemental Material:
The supplemental files are great, and I congratulate the authors on how well they’ve organised these. I would add in subheadings to supplemental file 2 to help make clear which text corresponds to which figure.
Supplemental Figures: See my comment in the results section about reorganising and combining plots to reduce the overall number of figures and help highlight common trends and make comparisons across the different aspects of jaw morphology. I’d also place jaw silhouettes in all the supplemental morphospace figures as done in Figures 6 & 8, so readers have a clear idea of the shape changes across the PC axes. I also think it may be worthwhile creating morphospace plots that distinguish key clades at a lower taxonomic level (e.g. basal synapsids, therapsids, diadectomorphs), however, feel free to ignore this point if you prefer as it doesn’t directly relate to your key findings or arguments; it’s just a suggestion to enhance general interest in the morphospace patterns.

---

## Round 0.2 · Minor Revisions

· Academic Editor

Minor Revisions

The manuscript is nearly ready for publication. I agree with one of the reviewers who still considers that minor changes are needed, such as a further consolidation of the results text and figures. I kindly ask the authors to carefully review these comments and revise the manuscript accordingly.

·

Basic reporting

Basic reporting:
The present manuscript is improved, being much more concise and clearer. Some parts of the results section (line 286-332) would benefit from further revision to cut them down and improve readability, but I appreciate the authors response on wanting to give a thorough account, so this is not a critical issue, especially given they have the space to do so.

Line 24-26: I would rephrase the sentence “Shifting the primary medium of feeding from water to air is expected to require changes to tetrapod mandibular form and function”, as it’s a little awkwardly phrased. Perhaps something more concise and direct like, “Feeding in air as opposed to water required changes to tetrapod mandibular form and function”.
Line 286-332: I understand the authors viewpoint, but I think the present paragraph could still be made a little more concise to improve readability. Now that Table 3 identifies the taxa with the min and max value for each trait, I suggest just referring readers to the table for these details, leaving the text to focus on the taxa and patterns not highlighted in the table and figures. Cuting this section down a little more would be ideal.
Line 355: I’d rephrase “exemplify the disparity functional disparity of amniotes with respect to nonamniotes” as it’s a little ambiguous/doesn’t clearly indicate that amniotes have higher functional disparity than amniotes. Be explicit in stating which group has greater disparity.
Line 690-691: Thanks for the clarification and I see my mistake. I’d recommend rephrasing to “subclades within Therocephalia and Pareiasauria” to avoid similar confusion.
Supplementary Figures –
I appreciate the authors response to keeping most figures separate, I would still suggest they try to further consolidate the supplementary figures to help readers synthesise the different results, which can be difficult when spread across multiple figures - I recommend similarly combining the boxplots of supplementary figures 1-3 and 4-7 into 2 figures respectively. You could give each scenario alphabetical listing (e.g., A, B, C, for amniote grouping 1, 2, 3).

Experimental design

No Comment.

Validity of the findings

The present manuscript provides greater elaboration on temporal trends in disparity and geological events. I believe that a little more intepretation of how these events could have driven the recovered disparity patterns would bolster the discussion, but I leave it to the authors discretion on whether they would like to apply this change.
Overall, it's a very strong morphofunctional study, with interesting findings on the early diversification of terrestrial tetrapods.

Additional comments

The authors have largely followed my recommendations. I would suggest further consolidation of the results text and figures, but this is perhaps more a preference than a critical issue. Therefore, I support the publication of this manuscript following the minor revisions outlined. Well done to the authors and I look forward to seeing this published.

---

## Round 0.3 · accepted · Accept

· Academic Editor

Accept

As handling editor, I consider that the revised version has satisfactorily addressed the reviewers’ comments and is now ready for publication. The modifications introduced have significantly improved the manuscript, and I believe the authors have done excellent work.